# PIRN: PROTOTYPICAL-BASED INTRA-MODAL RECONSTRUCTION WITH NORMALITY COMMUNICATION FOR MULTIMODAL ANOMALY DETECTION

Yiting Li[1], Xulei Yang[1][†], Jing Zhang[2], Sichao Tian[3], Jingyi Liao[1], Fayao Liu[1][†]

[1]Institute for Infocomm Research, A*STAR, Singapore
[2]School of Computer Science and Engineering, Macao University of Science and Technology
[3]Institute of Medicinal Plant Development, Chinese Academy of Medical Sciences and Peking Union Medical College, China
{li_yiting, yang_xulei, liu_fayao}@a-star.edu.sg

## ABSTRACT

Unsupervised multimodal anomaly detection (MAD) aims to detect anomalies by using both RGB and 3D modalities. However, existing methods struggle in few-shot scenarios where the number of normal training samples is limited. Specifically, cross-modal alignment approaches fail to learn reliable correspondences from scarce normal data, whereas memory-based methods often misclassify unseen normal variations as anomalies. To address these issues, we propose **PIRN**, a prototype-driven reconstruction framework equipped with explicit cross-modal knowledge transfer. Instead of relying on dense feature alignment or heavy memory banks, **PIRN** uses a compact set of learnable prototypes to capture diverse normal patterns and constrain feature reconstruction. Specifically, our framework incorporates three core innovations. We introduce Balanced Prototype Assignment (BPA), which employs balanced optimal transport to ensure uniform prototype utilization and prevent codebook collapse. Next, we propose Adaptive Prototype Refinement (APR), which uses gated prototype updates to dynamically expand the model's knowledge of unseen normal variations during inference. To enable each modality to assist the other in reconstructing, we further develop a Multimodal Normality Communication (MNC) module that exchanges high-level normal cues between modalities via gated cross-attention. Extensive experiments on the MVTec 3D-AD, Eyecandies, and Real-IAD benchmarks validate the effectiveness of **PIRN**, where it consistently achieves superior performance compared to existing baselines under challenging few-shot settings.

## 1 INTRODUCTION

Multimodal anomaly detection (MAD; Wang et al., 2023; Costanzino et al., 2024; Long et al., 2025) identifies anomalies by inspecting both RGB images and 3D point clouds. Compared with single-modality methods, MAD-based methods can reveal defects that are invisible to either modality alone. However, existing MAD methods still struggle in few-shot scenarios where only a handful of normal samples per class are available for training (Fang et al., 2023; Tian et al., 2024; Huang et al., 2022). For example, cross-modal alignment approaches such as CFM (Costanzino et al., 2024) and LSFA (Tu et al., 2024) attempt to learn dense correspondences between RGB and 3D modalities using only normal data. An anomaly is identified when features of one modality cannot be predicted from those of the other modality. However, with very few normal samples, the learned mapping captures only narrow cross-modal correlations and fails on unseen yet normal correspondences at test time. Memory-bank methods such as M3DM (Wang et al., 2023) and SG-DM (Chu et al., 2023) store normal features as exemplars and detect anomalies by measuring the divergence between a test sample and its k-nearest exemplars. With limited normal samples, memory-based models fail to capture the full range of normal variation, leading to false positives on mildly deviating test samples.

---

[†]Corresponding author.

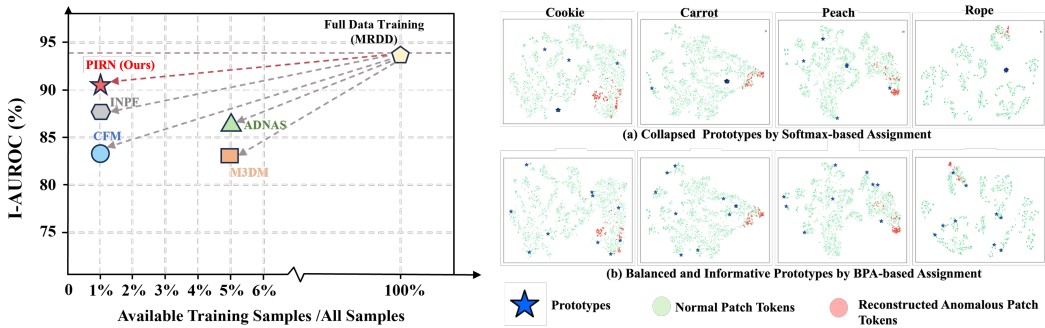

Figure 1: **Left:** Comparison with state-of-the-art methods on the Eyecandies dataset ($AUROC_I$ metric). PIRN achieves superior anomaly detection accuracy using less than 1% of the training data, significantly outperforming existing methods in data-scarce scenarios. **Right:** t-SNE visualization of patch tokens and prototypes in the RGB decoder feature space (MVTec 3D-AD, 10-shot setting). BPA (*bottom*) yields a more uniform prototype distribution over normal features, whereas a softmax assignment (*top*) results in underutilized/collapsed prototypes.

As such, both alignment- and memory-based approaches degrade significantly in the challenging few-shot settings (see Fig. 1 **Left**).

We address these limitations with **PIRN**: **P**rototypical-based **I**ntra-modal **R**econstruction with **N**ormality Communication for few-shot MAD. Rather than overfitting to sparse data via dense cross-modal matching or relying on large memory banks, **PIRN** emphasizes robust **Intra-modal Feature Reconstruction** using a vector-quantized codebook of discrete normality-aware prototypes (Van Den Oord et al., 2017). By reconstructing the features of each modality from a compact codebook, **PIRN** enforces an information bottleneck (Alemi et al., 2017; Seo et al., 2023; Zhang et al., 2024b) that retains only essential patterns of normal texture and geometry while ignoring irrelevant details. Consequently, anomalies that cannot be well represented by the prototypes yield large reconstruction errors.

Prototype-based reconstruction in a few-shot scenario faces several challenges. First, standard soft-assignment schemes (e.g., softmax attention) are prone to the codebook collapse issue (Zheng & Vedaldi, 2023). During training, a few prototypes tend to encode common normal modes, whereas the rest receive fewer updates and become inactive. This issue not only reduces the model's capacity but also narrows the prototype codebook's coverage of normality. Second, a static prototype codebook learned from scarce training data often fails to capture wide variations in normality (Zhang et al., 2024a; Wei et al., 2023). As a result, a normal test sample may contain unseen yet still-normal patterns that cannot align with any learned prototype, leading to false-positive predictions. Third, existing prototype-based AD methods (Mao et al., 2025; Luo et al., 2025) treat each modality in isolation, ignoring the complementary information between texture and geometry. Without effective cross-modal collaboration, subtle defects unique to one modality may go undetected.

We introduce three designs to address the above limitations. **First**, Balanced Prototype Assignment (BPA) formulates the patch-to-prototype matching task as a balanced optimal transport problem (Peyré & Cuturi, 2019), thus ensuring that each prototype captures a distinct normal pattern. This prevents codebook collapse and promotes diverse normal coverage with limited normal samples. As shown in Fig. 1 **Right**, BPA yields a more uniform prototype distribution than softmax assignment. **Second**, Adaptive Prototype Refinement (APR) bridges the train–test distribution gap by treating the prototypes as adaptive memory at inference. APR uses a lightweight GRU to update the prototype vectors based on the test image's normal context, without corrupting them with anomaly contexts. This on-the-fly refinement expands the prototypes' coverage to new normal variations that are absent during training. **Third**, we introduce Multimodal Normality Communication (MNC) that exchanges prototypical normality knowledge across modalities via a two-stage process. The first stage aligns high-level normal concepts encoded by prototypes across modalities through graph refinement. In the second stage, these refined prototypes serve as anchors to guide fine-grained feature reconstruction via cross-attention. As such, this allows each modality to reinforce the other's understanding of

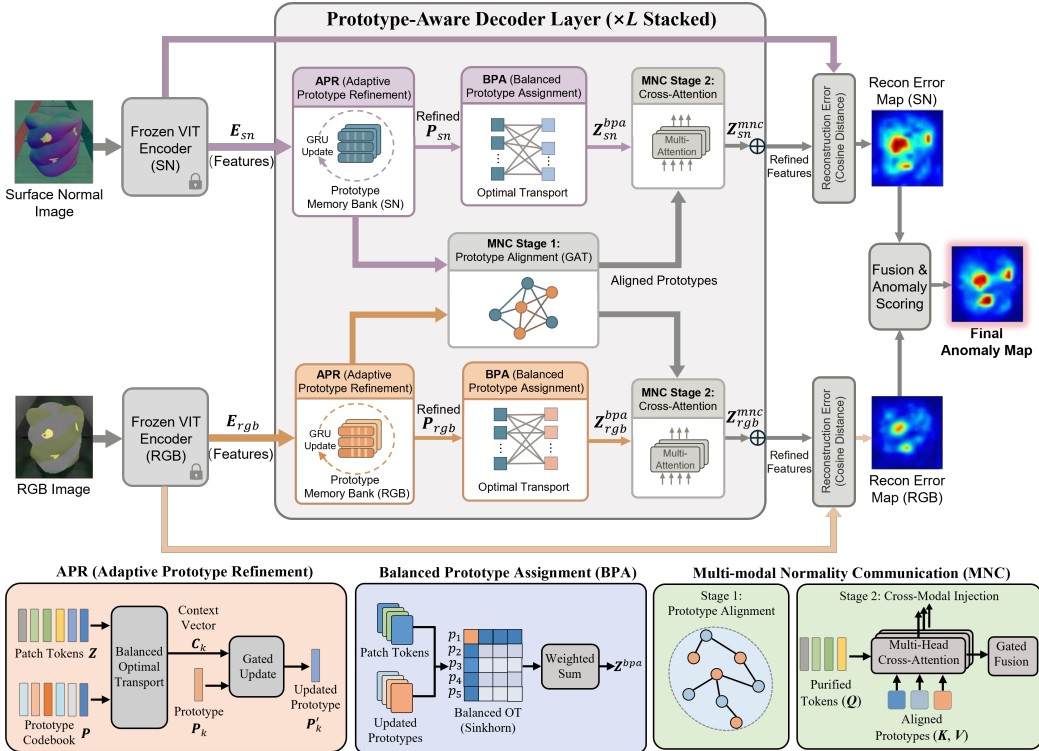

Figure 2: (a) **Overview of PIRN.** Given an RGB image and a surface-normal map, PIRN uses pre-trained frozen encoders to extract features $E_{rgb}$ and $E_{sn}$. A prototype-aware multi-layer decoder then reconstructs these features into $\mathbf{Z}^{\text{bpa}}$ (intra-modal purified) and $\mathbf{Z}^{\text{mnc}}$ (cross-modal purified), which are used to generate anomaly maps. PIRN introduces three key components: 1) APR for adaptive prototype refinement to capture unseen normal patterns at test time; 2) BPA for balanced prototype assignment to mitigate codebook collapse; and 3) MNC for cross-modal prototype communication. (b) Details of the three components.

normality, enabling more discriminative detection of challenging anomalies (e.g., subtle defects) that might go undetected when each modality is used in isolation.

Together, these modules enable our model to learn and communicate normal patterns effectively across modalities, significantly improving anomaly detection in data-scarce settings. Our main contributions are summarized as follows:

- We present **PIRN** – a robust *Prototypical-based Intra-modal Reconstruction with cross-modal Normality Communication* framework for few-shot MAD.

- We introduce BPA to prevent codebook collapse and capture more diverse normal patterns. A lightweight APR module is further proposed to expand the prototypes' coverage to unseen yet normal variations at inference.

- We propose an MNC mechanism that shares normal information across modalities via cross-modal knowledge transfer, enabling each modality to help reconstruct the other's normal features and clearly highlight anomalies.

## 2 RELATED WORK

**2D Anomaly Detection.** Many recent 2D anomaly detection (AD) methods constrain normal feature representations by using discrete prototypes to encode "normality." For example, HVQ-Trans (Lu et al., 2023) preserves typical normal patterns as a vector-quantized prototype codebook, preventing the "identical shortcut" issue and ensuring anomalies cannot be perfectly reconstructed. Similarly,

RLR (He et al., 2024) introduces a learnable reference representation to discourage shortcut solutions and explicitly model normal patterns. DPDL (Wang et al., 2025) learns multiple Gaussian prototypes and diffuses normal samples toward these cluster centers, forming a compact normal feature space to exclude anomalies. INP-Former (Luo et al., 2025) extracts intrinsic normal prototypes directly from each test image, eliminating reliance on external memory bank and achieving state-of-the-art performance in 2D AD tasks. Gong et al. (2019) introduce MemAE, a memory-augmented autoencoder that utilizes an explicit memory bank to record prototypical normal patterns, thereby constraining reconstruction to learned normality. Guo et al. (2023) propose a template-guided approach, utilizing exemplars from the normal training library to guide the hierarchical restoration of input features, detecting anomalies via reconstruction deviations. However, lacking explicit cross-modal interaction, such methods are not directly applicable to MAD tasks.

**Multimodal Anomaly Detection.** Existing MAD methods mostly rely on cross-modal alignment or memory banks, with some exploring architecture search and distillation. Cross-modal alignment approaches (e.g., CFM (Costanzino et al., 2024), LSFA (Tu et al., 2024)) learn to align RGB and 3D features using only normal data, detecting anomalies when one modality's features cannot be predicted by the other. These methods fuse texture and geometry cues effectively but need diverse normal samples to establish reliable cross-modal correspondences. Alternatively, memory-based models such as M3DM (Wang et al., 2023) and SG-DM (Chu et al., 2023) store normal feature patterns (either fused or modality-specific) and identify deviations as anomalies. Such methods suffer in few-shot settings: any unseen yet normal pattern not shown in the memory can lead to misidentification. Beyond alignment and memory methods, 3D-ADNAS (Long et al., 2025) optimizes feature fusion architectures via neural architecture search.

## 3 METHOD

### 3.1 FRAMEWORK OVERVIEW

To the best of our knowledge, **PIRN** (overview in Fig. 2) is the first multimodal anomaly detection (AD) framework to integrate a vector-quantized prototype codebook into a Vision Transformer (ViT; Dosovitskiy et al., 2021) encoder–decoder architecture. Specifically, for each modality, we learn a compact codebook of $K$ vector-quantized discrete prototypes. These prototypes serve as reference points for typical normal textures and geometries, constraining reconstruction to rely solely on normal information.

**Frozen ViT Encoder.** We employ two parallel ViT encoders, $\mathcal{E}_{\text{rgb}}$ and $\mathcal{E}_{\text{sn}}$, which are pre-trained and kept frozen. $\mathcal{E}_{\text{rgb}}$ processes the input RGB image, while $\mathcal{E}_{\text{sn}}$ processes the corresponding surface-normal map. We extract multi-scale features from a set of intermediate layers of each encoder and aggregate them via element-wise averaging to form a single feature map per modality (denoted $E_{rgb}$ and $E_{sn}$, each in $\mathbb{R}^{N \times C}$). These aggregated feature maps serve as both the input to the decoder and the target for reconstruction.

**Cascaded Prototype-Aware Decoder.** The decoder consists of a stack of prototype-aware layers that progressively reconstruct a normal version of input features. Each decoder layer performs three sequential operations. First, Adaptive Prototype Refinement (APR) updates each modality's prototype codebook via a gated recurrent unit (GRU; Chung et al., 2014), enhancing adaptability to the current sample. Next, Balanced Prototype Assignment (BPA) assigns each patch token to the updated prototypes via balanced optimal transport, promoting uniform prototype utilization. Finally, Multimodal Normality Communication (MNC) aligns the refined prototypes from both modalities through graph-based attention, and then exchanges high-level normality knowledge between the two modalities.

### 3.2 BALANCED PROTOTYPE ASSIGNMENT (BPA)

Allowing each token to softly match against all $K$ prototypes can lead to a codebook collapse: some prototypes may eventually become under-utilized, reducing the diversity of normal patterns the codebook can represent. BPA addresses this issue by formulating the token-to-prototype assignment as a balanced *optimal transport* (OT) problem. Instead of using softmax assignment that might

over-concentrate on a few prototypes, BPA promotes two properties for a more uniform prototype usage: (1) **patch-to-prototype selectivity**, encouraging each patch token to concentrate its mass on only a few prototype codes (controlled by the OT cost and entropic regularization); and (2) **uniform prototype utilization**, ensuring all prototypes receive a balanced share of patch assignments. Therefore, BPA encourages each prototype to specialize in a distinct normal pattern, yielding a more diverse and representative codebook.

Specifically, let $\mathbf{Z} = \{z_n\}_{n=1}^N$ denote the set of $N$ patch tokens input to a given decoder layer (for the first decoder layer of each modality, $\mathbf{Z}$ equals the corresponding encoder output $E_{rgb}$ or $E_{sn}$). Let $P = \{p_k\}_{k=1}^K$ denote the prototype vectors of a specific modality's codebook. In practice, before applying BPA we first refine the prototypes using APR (detailed in the next section), which adapts $P$ to the normal context of the input image. This ensures that BPA operates on prototypes already tailored to the current sample.

We then define a cost matrix $C \in \mathbb{R}^{N \times K}$ with $C_{nk} = 1 - \frac{z_n \cdot p_k}{\|z_n\|\|p_k\|}$ representing the cosine distance between patch token $z_n$ and prototype $p_k$. BPA aims to find an optimal transport plan $T^* \in \mathbb{R}_{\geq 0}^{N \times K}$ that minimizes the assignment cost under equal-mass constraints:

$$T^* = \arg\min_T \sum_{n=1}^N \sum_{k=1}^K T_{nk}\, C_{nk} \tag{1}$$

$$\text{s.t.} \quad T\mathbf{1}_K = \mathbf{a}, \quad T^\top \mathbf{1}_N = \mathbf{b},$$

where $\mathbf{a} = \mathbf{1}_N$ and $\mathbf{b} = \frac{N}{K}\mathbf{1}_K$.

This optimal transport formulation yields a balanced soft assignment to avoid trivial solutions (e.g., all patches assigned to a single prototype) and ensures full prototype utilization. We solve it using the Sinkhorn algorithm (Cuturi, 2013) with entropic regularization. We then use the optimal plan $T^*$ to reconstruct each patch token as a weighted combination of these prototypes.

$$z_n^{\text{bpa}} = \sum_{k=1}^K T_{nk}^* p_k. \tag{2}$$

This effectively projects the input query tokens $\{z_n\}_{n=1}^N$ onto the prototype space under the learned OT weights $T^*$. Thus, BPA acts as an information bottleneck by reconstructing each patch token from only a limited set of normality-aware prototypes. As a result, only normal patterns can be reliably reconstructed, and anomalous features in the query input are largely filtered out, leading to large reconstruction errors in anomalous regions. We refer to $\mathbf{Z}^{\text{bpa}} = \{z_n^{\text{bpa}}\}_{n=1}^N$ as *intra-modal reconstruction*, since they are derived solely from the normal prototypes of the same modality.

### 3.3 ADAPTIVE PROTOTYPE REFINEMENT (APR) VIA OPTIMAL TRANSPORT

Our framework relies on a set of prototypes $\{p_k\}_{k=1}^K$ as a compact codebook of normal patterns, which allows the model to capture diverse normal patterns during training and adapt to unseen variations at test time. To achieve this, we introduce Adaptive Prototype Refinement (APR), which dynamically refines the prototypes using the normal context extracted from the current input. Importantly, APR operates on the patch tokens $\mathbf{Z}$ that are extracted from the previous decoder layer (or the encoder's output for the first decoder layer), before any reconstruction is performed in the current decoder layer.

To ensure that only normal patch tokens can contribute to each prototype, we compute an optimal transport alignment between the patch tokens and the prototypes. Similar to Eq. (1), we derive the OT plan $\Gamma^*$. This plan associates each prototype $p_k$ with a weighted subset of patch tokens in $\mathbf{Z} = \{z_n\}_{n=1}^N$ that it best represents. We then compute a context vector for prototype $p_k$ as a (column-normalized) OT-weighted mean of its matched patch tokens: $c_k = \sum_{n=1}^N \bar{\Gamma}_{nk}^* z_n$, where $\bar{\Gamma}_{nk}^*$ denotes the $k$-th column of $\Gamma^*$ normalized such that $\sum_{n=1}^N \bar{\Gamma}_{nk}^* = 1$.

This OT-based context extraction provides robust guidance for prototype refinement. By finding an optimal matching between prototypes and patch tokens under balanced constraints, an out-of-distribution (anomalous) patch tends be assigned more diffusely across prototypes (i.e., with low affinity to any single prototype), thereby contributing weakly to each prototype context. This

encourages each prototype $p_k$ to be updated primarily using reliable in-distribution (normal) patches, enabling robust refinement even in the presence of minor anomalies. Finally, we update each prototype $p_k$ to $p'_k$ by incorporating its context $c_k$ via a GRU, whose gating mechanism restricts the integration of unreliable anomalous contexts by leaving $p_k$ essentially unchanged.

## 3.4 MULTIMODAL NORMALITY COMMUNICATION (MNC)

To model complementary cues from texture (RGB) and geometry (surface normals), we introduce a Multimodal Normality Communication (MNC) module that exchanges normal information between the two branches. The key idea is that each modality can assist the other in understanding normality, thereby better highlighting true anomalies and suppressing false positives. To ensure robust knowledge transfer, MNC exchanges prototype-based normal knowledge between modalities, rather than raw patch features that may contain anomalies during testing. The decoder of each modality is guided to reconstruct features not only from its own prototypes but also from high-level normal patterns of the other modality. MNC operates in two stages: a *prototype alignment* stage and a *cross-modal normality injection* stage.

**Stage 1: 2D and 3D Prototype Alignment.** We treat all prototypes from both modalities as nodes in a unified graph and perform cross-modal message passing to align them. Specifically, we construct a graph with $2K$ nodes, consisting of $K$ RGB prototypes and $K$ surface-normal prototypes. We connect each prototype to its nearest neighbors in the other modality using KNN in the feature space of $\ell_2$-normalized prototypes, and then apply a multi-head Graph Attention Network (GAT; Veličković et al., 2018) to propagate information across these edges. This graph-based refinement pulls the two sets of prototypes into a shared semantic space: prototypes representing similar structures (e.g., a flat surface or an edge) are drawn closer and enriched with complementary context from the other modality. Let $\mathbf{P}'_{rgb}$ and $\mathbf{P}'_{sn}$ denote the refined RGB and surface-normal prototype sets after this alignment. As a result, the two branches obtain *aligned prototypes* that encode a consistent cross-modal notion of normal texture and geometry. Similar prototype-level alignment strategies (Huang et al., 2025; Tang et al., 2023; Pahde et al., 2021) have proven effective in multimodal representation learning.

**Stage 2: Cross-Modal Normality Injection.** After alignment, the refined prototypes serve as anchors to guide fine-grained feature reconstruction via cross-attention. In this stage, each patch token from one modality will attend to the other modality's refined prototypes to inject any normal information it lacks. To suppress anomalous details in test samples, we first purify each modality's patch tokens using its intra-modal information. Specifically, we use the intra-modal purified tokens $z_n^{\mathrm{bpa}}$ as an attention mask to reweight the original patch tokens $z_n$ channel-wise. This yields purified tokens $\mathbf{Z}' = \{z_n \cdot \sigma(z_n^{\mathrm{bpa}})\}_{n=1}^N$, where $\sigma(\cdot)$ is the sigmoid function. These purified tokens $\mathbf{Z}'$ are then used as queries in the cross-modal attention.

For cross-modal knowledge exchange, we employ a cross-attention layer (Vaswani et al., 2017) where the refined prototypes of one modality act as keys and values, and the purified patch tokens from the other modality act as queries. Taking the RGB branch as an example, let $\mathbf{Z}'_{rgb}$ denote the purified tokens of the RGB branch and $\mathbf{P}'_{sn}$ denote the set of stage-1 refined prototypes from the surface-normal branch. We compute the cross-attention output as:

$$\mathrm{CA}(\mathbf{Z}'_{rgb}, \mathbf{P}'_{sn}) = \mathrm{SoftMax}\left(\frac{\mathbf{Z}'_{rgb} W_Q (\mathbf{P}'_{sn} W_K)^\top}{\sqrt{d}}\right)(\mathbf{P}'_{sn} W_V), \tag{3}$$

where $W_Q, W_K, W_V$ are projection matrices and $d$ is the channel dimension of $\mathbf{Z}'_{rgb}$.

To prevent overwhelming the patch features with irrelevant information, we introduce a learnable gating scalar $\gamma$ to modulate the cross-attention output. Specifically, we add a scaled version of the cross-attention result to the original token representation:

$$\begin{aligned}
\mathbf{Z}^{\mathrm{mnc}}_{rgb} &= \mathbf{Z}'_{rgb} + g_{rgb} \cdot \mathrm{CA}(\mathbf{Z}'_{rgb}, \mathbf{P}'_{sn}), & g_{rgb} &= \tanh(\gamma_{rgb}), \\
\mathbf{Z}^{\mathrm{mnc}}_{sn} &= \mathbf{Z}'_{sn} + g_{sn} \cdot \mathrm{CA}(\mathbf{Z}'_{sn}, \mathbf{P}'_{rgb}), & g_{sn} &= \tanh(\gamma_{sn}),
\end{aligned} \tag{4}$$

where $\gamma_{rgb}$ and $\gamma_{sn}$ are learnable scalar parameters (one per decoder layer) and $g_{rgb}, g_{sn}$ serve as gates on the cross-modal information. This gating mechanism allows the network to control the extent

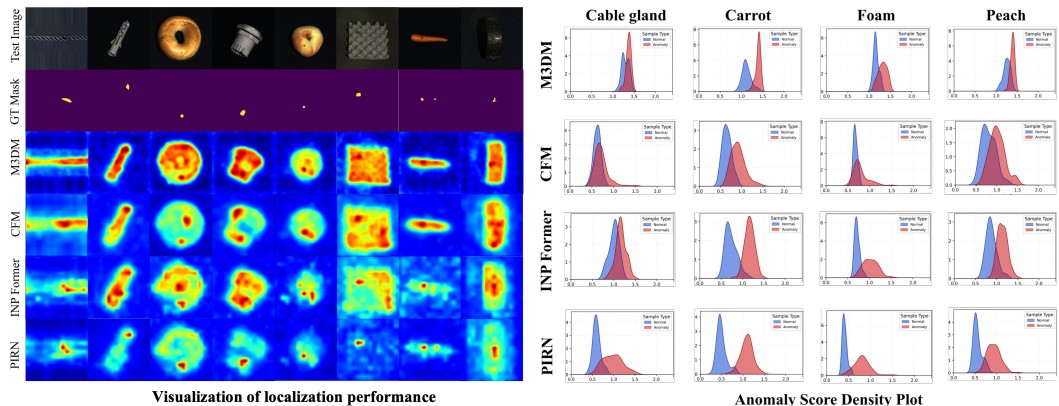

Figure 3: **Left:** Compared to existing MAD methods (10-shot), our anomaly maps are sharper with fewer false positives. **Right:** Comparison of anomaly score distributions for normal and anomalous samples (10-shot, MVTec-3D-AD). **PIRN** shows clearer distribution separation.

of cross-modal fusion for each layer. By exchanging high-level normality knowledge and injecting it into fine-grained patch tokens, MNC establishes a robust correspondence between modalities at the prototype level. Unlike methods that attempt dense patch-to-patch alignment between modalities (which can be unreliable given limited data), our prototype-centric exchange avoids direct dense mappings and thus offers greater robustness on unseen test samples.

We refer to $\mathbf{Z}_{rgb}^{mnc}$ and $\mathbf{Z}_{sn}^{mnc}$ as *cross-modal purified reconstructions*, as they are obtained using normal prototypes from both modalities. We then fuse $\mathbf{Z}^{bpa}$ and $\mathbf{Z}^{mnc}$ via element-wise summation to produce the final reconstructed features for each modality: $\mathbf{Z}_{rgb}^{rec} = \mathbf{Z}_{rgb}^{bpa} + \mathbf{Z}_{rgb}^{mnc}$ and $\mathbf{Z}_{sn}^{rec} = \mathbf{Z}_{sn}^{bpa} + \mathbf{Z}_{sn}^{mnc}$.

**Training and Inference** We train **PIRN** end-to-end using an intra-modal feature reconstruction loss, e.g., a soft mining loss (Luo et al., 2025). In practice, we minimize the cosine distance between the encoder's patch embeddings ($E_{rgb}$ and $E_{sn}$) and the corresponding reconstructed embeddings ($\mathbf{Z}_{rgb}^{rec}$ and $\mathbf{Z}_{sn}^{rec}$) across all spatial locations for both modalities. During inference, we compute an anomaly score map by comparing encoder-extracted features with reconstructed features. For modality $m \in \{rgb, sn\}$ and the $i$-th patch, the anomaly score is $d_i^{(m)} = 1 - \cos(E_i^{(m)}, \mathbf{Z}_i^{rec,(m)})$. These patch-level anomaly maps are upsampled to the input resolution and summed across branches to produce a fused anomaly heatmap. The final image-level anomaly score is the maximum value in this fused heatmap.

## 4 EXPERIMENTS

**Implementation Details.** We adopt a ViT-Base / 14 transformer as the backbone encoder for RGB and surface-normal inputs, initialized with DINOv2 (Oquab et al., 2023) pre-trained weights and kept **frozen** during training. We follow FIND's (Li et al., 2025) procedure to generate surface normal maps from 3D point clouds. To obtain robust multi-scale features, we aggregate patch tokens from layers 2–10 of the pretrained ViT by element-wise averaging. Unless otherwise specified, we use $K = 10$ prototypes per modality. The decoder is a cascaded architecture with $L = 2$ layers. We use Adam (Kingma & Ba, 2014) optimizer (learning rate $1 \times 10^{-4}$) for 60 epochs in few-shot tasks and 8 epochs in all-shot tasks.

**Baselines and Evaluation Protocol.** We compare PIRN with representative MAD baselines, including BTF (Horwitz & Hoshen, 2023), AST (Rudolph et al., 2023), M3DM (Wang et al., 2023), CFM (Costanzino et al., 2024), and 3D-ADNAS (Long et al., 2025). We also include the recent 2D prototype-based reconstruction method INP-Former (Luo et al., 2025). To ensure a fair comparison in the multimodal setting, the 2D INP-Former baseline is adapted to a two-stream architecture that

| k-Shot | Method | MVTec-3D-AD | | | Eyecandies | | |
|---|---|---|---|---|---|---|---|
| | | $\text{AUROC}_I$ | $\text{AUROC}_P$ | AUPRO | $\text{AUROC}_I$ | $\text{AUROC}_P$ | AUPRO |
| 5 | BTF (Horwitz & Hoshen, 2023) | 0.671 | 0.980 | 0.920 | 0.652 | 0.815 | 0.738 |
| | AST (Rudolph et al., 2023) | 0.680 | 0.950 | 0.903 | 0.633 | 0.741 | 0.691 |
| | M3DM (Wang et al., 2023) | 0.822 | 0.984 | 0.937 | 0.764 | 0.871 | 0.807 |
| | CFM (Costanzino et al., 2024) | 0.811 | 0.986 | 0.949 | 0.795 | 0.879 | 0.801 |
| | 3D-ADNAS (Long et al., 2025) | 0.826 | – | – | 0.775 | 0.875 | – |
| | INP-Former (Luo et al., 2025) | 0.851 | 0.988 | 0.957 | 0.859 | 0.946 | 0.862 |
| | **Ours** | **0.890** | **0.990** | **0.960** | **0.895** | **0.955** | **0.887** |
| 10 | BTF (Horwitz & Hoshen, 2023) | 0.695 | 0.983 | 0.928 | 0.685 | 0.834 | 0.806 |
| | AST (Rudolph et al., 2023) | 0.689 | 0.946 | 0.835 | 0.671 | 0.767 | 0.624 |
| | M3DM (Wang et al., 2023) | 0.845 | 0.986 | 0.943 | 0.824 | 0.890 | 0.812 |
| | CFM (Costanzino et al., 2024) | 0.845 | 0.987 | 0.954 | 0.838 | 0.903 | 0.825 |
| | 3D-ADNAS (Long et al., 2025) | 0.848 | – | – | 0.807 | 0.869 | – |
| | INP-Former (Luo et al., 2025) | 0.885 | 0.989 | 0.960 | 0.872 | 0.947 | 0.870 |
| | **Ours** | **0.922** | **0.991** | **0.966** | **0.912** | **0.969** | **0.896** |
| 50 | BTF (Horwitz & Hoshen, 2023) | 0.806 | 0.989 | 0.947 | 0.721 | 0.856 | 0.824 |
| | AST (Rudolph et al., 2023) | 0.794 | 0.974 | 0.929 | 0.739 | 0.862 | 0.715 |
| | M3DM (Wang et al., 2023) | 0.907 | 0.989 | 0.955 | 0.836 | 0.933 | 0.846 |
| | CFM (Costanzino et al., 2024) | 0.906 | 0.991 | 0.965 | 0.852 | 0.926 | 0.851 |
| | 3D-ADNAS (Long et al., 2025) | 0.890 | – | – | 0.868 | 0.912 | – |
| | INP-Former (Luo et al., 2025) | 0.921 | 0.991 | 0.965 | 0.902 | 0.967 | 0.892 |
| | **Ours** | **0.945** | **0.993** | **0.970** | **0.924** | **0.975** | **0.908** |
| All | BTF (Horwitz & Hoshen, 2023) | 0.865 | 0.992 | 0.959 | 0.740 | 0.883 | 0.845 |
| | AST (Rudolph et al., 2023) | 0.937 | 0.976 | 0.944 | 0.780 | 0.902 | 0.744 |
| | M3DM (Wang et al., 2023) | 0.945 | 0.992 | 0.964 | 0.882 | 0.977 | 0.887 |
| | CFM (Costanzino et al., 2024) | 0.954 | 0.993 | 0.971 | 0.881 | 0.974 | 0.887 |
| | 3D-ADNAS (Long et al., 2025) | 0.951 | – | – | 0.946 | 0.970 | – |
| | INP-Former (Luo et al., 2025) | 0.952 | 0.994 | 0.971 | 0.934 | 0.981 | 0.918 |
| | **Ours** | **0.963** | **0.994** | **0.973** | **0.948** | **0.983** | **0.923** |

Table 1: Comparison of anomaly detection and localization performance on **MVTec-3D-AD** and **Eyecandies** under different training shots.

| Modules | | | | Metrics | | |
|---|---|---|---|---|---|---|
| BPA | APR | MNC | | $\text{AUROC}_I$ | $\text{AUROC}_P$ | AUPRO |
| ✗ | ✗ | ✗ | | 0.828 | 0.976 | 0.952 |
| ✗ | ✓ | ✓ | | 0.883 | 0.990 | 0.956 |
| ✓ | ✗ | ✓ | | 0.916 | 0.990 | 0.961 |
| ✓ | ✓ | ✗ | | 0.867 | 0.988 | 0.947 |
| ✓ | ✓ | ✓ | | 0.922 | 0.991 | 0.966 |

Table 2: Contribution of each component in few-shot regime, performance under 10-shot normal setting on **MVTec-3D-AD**.

| Setting / Modality | $\text{AUROC}_I$ | $\text{AUROC}_P$ | AUPRO | $\text{F1max}_I$ |
|---|---|---|---|---|
| **5-shot** | | | | |
| RGB-only | 0.794 | 0.966 | 0.890 | 0.910 |
| Surface Normals-only | 0.854 | 0.972 | 0.912 | 0.932 |
| **RGB + Surface Normals** | 0.890 | 0.990 | 0.960 | 0.940 |
| **10-shot** | | | | |
| RGB-only | 0.827 | 0.968 | 0.901 | 0.921 |
| Surface Normals-only | 0.879 | 0.974 | 0.920 | 0.937 |
| **RGB + Surface Normals** | 0.922 | 0.991 | 0.966 | 0.951 |
| **All-shot** | | | | |
| RGB-only | 0.874 | 0.977 | 0.917 | 0.929 |
| Surface Normals-only | 0.937 | 0.977 | 0.928 | 0.946 |
| **RGB + Surface Normals** | 0.963 | 0.994 | 0.973 | 0.962 |

Table 3: Effect of modality availability (RGB-only vs. SN-only vs. RGB+SN) under different training shots on **MVTec-3D-AD**.

processes RGB images and surface normal maps independently. Both branches use the same ViT-B/14 backbone and input resolution as in PIRN. Following the same post-processing (upsampling and Gaussian smoothing) used in our pipeline, their patch-level anomaly maps are fused via element-wise summation to produce the final predictions. We report AUROC at the image-level ($\text{AUROC}_I$), pixel-level AUROC ($\text{AUROC}_P$), and AUPRO following prior work (Costanzino et al., 2024).

**Main Results.** As shown in Tab. 1, our method consistently outperforms the best-performing baseline on both MVTec-3D-AD and Eyecandies across all metrics in varying few-shot settings. On MVTec-3D-AD, it improves $\text{AUROC}_I$ by +3.9 (5-shot), +3.7 (10-shot), and +2.4 (50-shot) over the strongest baseline. On Eyecandies, it improves $\text{AUROC}_I$ by +3.6 (5-shot), +4.0 (10-shot), and +2.2 (50-shot). These consistent gains validate PIRN's effectiveness under extremely limited training data. PIRN also achieves the best performance in the full-shot setting.

**Qualitative Analysis.** As shown in Fig. 3, **PIRN** enables a more accurate anomaly localization with fewer false positives. Furthermore, our method yields a more separable anomaly score distribution with a larger margin and less overlap between normal and anomalous scores. These qualitative results further validate **PIRN**'s effectiveness in few-shot scenarios. Fig. 5 in the appendix further highlights the complementary strengths of the RGB and 3D branches. For texture-only anomalies, the RGB branch alone accurately localizes the defect region, whereas the surface-normal branch can produce false positives. We further report per-category results in Appendix Tab. 11.

**Ablation of Token Aggregation in APR.** Tab. 7 compares various token aggregation methods used by APR. Global averaging performs the worst ($\text{AUROC}_I$ 91.5%) due to indiscriminate token pooling. Top-$k$ averaging improves performance ($\text{AUROC}_I = 92.1\%$), whereas balanced OT yields the best results. The slight gain of balanced OT over top-$k$ averaging indicates that balanced token contributions enable more consistent prototype learning.

**Effect of Modality Availability/Each Component** Tab. 3 reports single-modality detection results. Across all shot settings, the model with surface normals (SN) consistently outperforms RGB, e.g., achieving 0.879 vs. 0.827 $\text{AUROC}_I$ at 10 shots. This indicates that geometric cues carry richer discriminative information for anomaly detection. Combining both modalities yields further improvement, with the performance gain most pronounced under the 5-shot setting, where each modality has limited coverage of normal variation. This confirms that MNC's cross-modal communication is particularly valuable when per-modality representations are underrepresented due to scarce training data.

| Method | AUROC$_I$ ↑ | FLOPs↓ (G) | Latency↓ (ms) |
|---|---|---|---|
| M3DM (2023) | 0.845 | 263.27 | 21.61 |
| CFM (2024) | 0.845 | 552.53 | 60.57 |
| FIND (SOTA) | 0.921 | 728.46 | 76.09 |
| **PIRN (Ours)** | **0.922** | **103.36** | **17.49** |

| Prototypes $K$ | AUROC$_I$ | AUROC$_P$ | AUPRO | F1max$_I$ |
|---|---|---|---|---|
| $K = 5$ | 0.954 | 0.993 | 0.969 | 0.957 |
| $K = 10$ | **0.963** | **0.994** | **0.973** | **0.962** |
| $K = 50$ | 0.940 | 0.991 | 0.966 | 0.952 |
| $K = 100$ | 0.901 | 0.984 | 0.950 | 0.947 |

Table 4: Computation efficiency comparison on 10-shot MVTec-3D-AD.

Table 5: Ablation on the number of prototypes $K$ in the all-shot setting on MVTec-3D-AD (normal-only).

| Number of layers | AUROC$_I$ | AUROC$_P$ | AUPRO |
|---|---|---|---|
| $L = 1$ | 0.921 | 0.993 | 0.967 |
| $L = 2$ | **0.924** | **0.994** | **0.968** |
| $L = 4$ | 0.913 | 0.992 | 0.964 |
| $L = 8$ | 0.869 | 0.985 | 0.949 |

| Method | AUROC$_I$ | AUROC$_P$ | AUPRO |
|---|---|---|---|
| w/o APR module | 0.916 | 0.990 | 0.961 |
| Global Averaging | 0.915 | 0.989 | 0.960 |
| Top-$k$ Averaging | 0.921 | 0.991 | 0.964 |
| Balanced Optimal Transport | 0.922 | 0.991 | 0.966 |

Table 6: Ablation on the number of decoder layers in the 10-shot setting (only normal data).

Table 7: Performance on **MVTec-3D-AD** under different aggregation methods in APR.

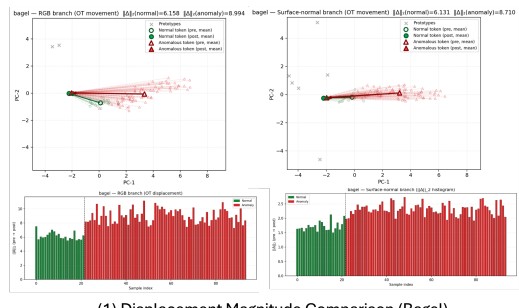
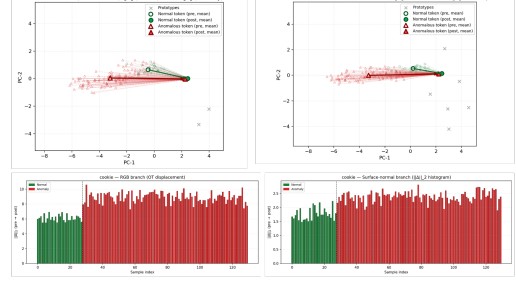

(1) Displacement Magnitude Comparison (Bagel)        (2) Displacement Magnitude Comparison (Cookie)

Figure 4: Visualization of Feature Displacement via BPA Routing. Green: normal; Red: anomalous.

We further validate each proposed module on the MVTec-3D-AD dataset, with results in Tab. 2. The baseline model (first row) excludes all proposed modules. The full **PIRN** model achieves superior performance. Removing each component from the full model results in a consistent performance drop, validating the contribution of every component.

**Effect of Codebook Size/Decoder Depth.** Tab. 5 examines the prototype count $K$ in the all-shot setting. With only $K = 5$ prototypes, the codebook is too small to cover the diversity of normal patterns. In contrast, a larger codebook size ($K = 50/100$) degrades performance because excessive prototypes weaken the information bottleneck: it allows anomalous patches to find close matches and be faithfully reconstructed. We also conduct comprehensive ablation studies on the number of decoder layers ($L$) under the 10-shot setting (Table 6), using only normal data. We find $L = 2$ provides the optimal trade-off (0.924 AUROC$_I$). Increasing the depth further (e.g., $L = 8$) leads to significant performance degradation. This demonstrates that excessively deep decoders suffer from overfitting in this few-shot regime.

**Computational Efficiency.** To clarify the trade-off between complexity and performance, we compare **PIRN** with representative reconstruction-based MAD baselines, including M3DM (Wang et al., 2023), CFM (Costanzino et al., 2024), and the recent SOTA FIND (Li et al., 2025), on the 10-shot MVTec-3D-AD setting. We report accuracy (AUROC$_I$) together with FLOPs and latency (input $224 \times 224$, batch size = 1). As shown in Tab. 4, PIRN achieves the best AUROC$_I$ (0.922) while being the most efficient, requiring only 103.36G FLOPs and 17.49ms latency, which is **85% fewer FLOPs** and **4.35× faster** than FIND (728.46G, 76.09ms).

**Analysis of Prototype-based Normality Encoding** To better interpret how PIRN's prototypes encode normality, we added a new OT-movement visualization in Fig. 4. For several MVTec-3D-AD categories (e.g., bagel, peach), we visualize the displacement of patch tokens from their initial

Table 8: Comparison of different methods on the Real-IAD D3 dataset. The highest value in each row is marked in red, and the second highest value in blue.

| Modality | RGB | | | | 3D | | 2D+3D | | | | | | D³ | | RGB + SN | |
|---|---|---|---|---|---|---|---|---|---|---|---|---|---|---|---|---|
| Model | Cflow | | SimpleNet | | PointMAE | | AST | | PointMAE+PatchCore | | M3DM | | D³M | | PIRN (Ours) | |
| Metrics | AUROC$_I$ | AUROC$_P$ | AUROC$_I$ | AUROC$_P$ | AUROC$_I$ | AUROC$_P$ | AUROC$_I$ | AUROC$_P$ | AUROC$_I$ | AUROC$_P$ | AUROC$_I$ | AUROC$_P$ | AUROC$_I$ | AUROC$_P$ | AUROC$_I$ | AUROC$_P$ |
| audio_jack_socket | 0.943 | 0.944 | 0.973 | 0.926 | 0.763 | 0.655 | 0.860 | 0.590 | 0.926 | 0.673 | 0.981 | 0.699 | 0.983 | 0.757 | 0.950 | 0.964 |
| common_mode_filter | 0.271 | 0.847 | 0.717 | 0.822 | 0.725 | 0.687 | 0.899 | 0.802 | 0.523 | 0.673 | 0.580 | 0.934 | 0.618 | 0.947 | 0.826 | 0.883 |
| connector_housing-female | 0.839 | 0.921 | 0.795 | 0.891 | 0.958 | 0.428 | 0.914 | 0.716 | 0.870 | 0.919 | 0.920 | 0.979 | 0.931 | 0.951 | 0.972 | 0.971 |
| crimp_st_cable_mount_box | 0.180 | 0.442 | 0.372 | 0.745 | 0.291 | 0.363 | 0.485 | 0.589 | 0.713 | 0.931 | 0.749 | 0.933 | 0.811 | 0.969 | 0.659 | 0.961 |
| dc_power_connector | 0.661 | 0.726 | 0.661 | 0.725 | 0.849 | 0.507 | 0.995 | 0.770 | 0.720 | 0.921 | 0.715 | 0.950 | 0.922 | 0.947 | 0.944 | 0.994 |
| ethernet_connector | 0.967 | 0.853 | 0.981 | 0.866 | 1.000 | 0.656 | 1.000 | 0.906 | 0.947 | 0.956 | 0.983 | 0.978 | 0.996 | 0.970 | 0.997 | 0.992 |
| ferrite_bead | 0.529 | 0.914 | 0.408 | 0.806 | 0.634 | 0.717 | 0.894 | 0.817 | 0.913 | 0.932 | 0.965 | 0.966 | 0.967 | 0.978 | 0.717 | 0.993 |
| fork_crimp_terminal | 0.462 | 0.657 | 0.416 | 0.945 | 0.422 | 0.62 | 0.595 | 0.773 | 0.769 | 0.952 | 0.780 | 0.964 | 0.819 | 0.946 | 0.978 | 0.991 |
| fuse_holder | 0.853 | 0.861 | 0.564 | 0.957 | 0.309 | 0.605 | 0.597 | 0.754 | 0.736 | 0.927 | 0.770 | 0.948 | 0.866 | 0.915 | 0.998 | 0.996 |
| headphone_jack_socket | 0.996 | 0.914 | 0.933 | 0.879 | 0.607 | 0.633 | 0.660 | 0.696 | 0.919 | 0.942 | 0.982 | 0.982 | 0.994 | 0.987 | 0.942 | 0.975 |
| humidity_sensor | 0.781 | 0.836 | 0.737 | 0.890 | 0.644 | 0.562 | 0.565 | 0.723 | 0.689 | 0.933 | 0.717 | 0.958 | 0.780 | 0.969 | 0.744 | 0.991 |
| knob_cap | 0.637 | 0.893 | 0.672 | 0.879 | 0.656 | 0.425 | 0.919 | 0.656 | 0.903 | 0.958 | 0.925 | 0.938 | 0.931 | 0.947 | 0.923 | 0.976 |
| lattice_block_plug | 0.833 | 0.852 | 0.790 | 0.898 | 0.769 | 0.776 | 0.842 | 0.919 | 0.911 | 0.923 | 0.917 | 0.958 | 0.939 | 0.941 | 0.892 | 0.969 |
| lego_pin_connector_plate | 0.828 | 0.877 | 0.857 | 0.947 | 0.361 | 0.482 | 0.847 | 0.629 | 0.662 | 0.759 | 0.681 | 0.734 | 0.891 | 0.889 | 0.981 | 0.980 |
| lego_propeller | 0.615 | 0.739 | 0.939 | 0.799 | 0.348 | 0.620 | 0.471 | 0.703 | 0.540 | 0.727 | 0.530 | 0.773 | 0.739 | 0.863 | 1.000 | 0.933 |
| limit_switch | 0.846 | 0.950 | 0.823 | 0.790 | 0.763 | 0.545 | 0.804 | 0.641 | 0.822 | 0.938 | 0.863 | 0.966 | 0.925 | 0.984 | 0.961 | 0.971 |
| miniature_lifting_motor | 0.402 | 0.799 | 0.402 | 0.760 | 0.717 | 0.435 | 0.766 | 0.467 | 0.948 | 0.962 | 0.975 | 0.991 | 0.823 | 0.961 | 0.604 | 0.838 |
| power_jack | 0.354 | 0.664 | 0.176 | 0.489 | 0.433 | 0.687 | 0.564 | 0.645 | 0.981 | 0.923 | 0.996 | 0.902 | 0.973 | 0.947 | 0.595 | 0.862 |
| purple_clay_pot | 0.343 | 0.571 | 0.343 | 0.938 | 0.869 | 0.271 | 0.635 | 0.445 | 0.921 | 0.961 | 0.944 | 0.953 | 0.962 | 0.922 | 0.871 | 0.997 |
| telephone_spring_switch | 0.575 | 0.910 | 0.627 | 0.916 | 0.771 | 0.413 | 0.951 | 0.551 | 0.827 | 0.944 | 0.856 | 0.936 | 0.934 | 0.957 | 0.904 | 0.987 |
| Avg | 0.645 | 0.808 | 0.659 | 0.843 | 0.644 | 0.554 | 0.693 | 0.650 | 0.812 | 0.905 | 0.841 | 0.922 | 0.890 | 0.937 | 0.873 | 0.961 |

feature space location ($z_{\text{pre}}$) to their final location after BPA+APR+MNC reconstruction ($z_{\text{post}}$). We project prototypes and tokens into a shared 2D PCA space and draw the displacement vectors ($\Delta = z_{\text{post}} - z_{\text{pre}}$). In the plots, translucent lines show per-token movements, and bold arrows indicate the average movement of normal (green) and anomalous (red) tokens. BPA encourages prototypes to serve as stable anchors for normal patterns. Normal embeddings that initially lie near prototype clusters undergo only small shifts during reconstruction. This suggests that the learned prototype codebook covers in-distribution normal patterns well. In contrast, anomalous tokens that initially lie farther away require larger displacements toward normal prototypes. The displacement histograms validates that our prototype-based reconstruction induces strong normal/anomalous discrimination.

**Experiment results on Real-IAD D3 dataset.** We further conduct comprehensive experiments on the challenging Real-IAD D3 (Zhu et al., 2025) dataset in the full-data training setting. Real-IAD D3 comprises industrial components in real-world settings with various anomaly types and complex geometries. As Table 8 shows, PIRN achieves highly competitive performance. It achieves the best overall anomaly localization (AUROC$_P$ 0.961) in 13 out of 20 categories, and it also achieves the second-best anomaly detection (AUROC$_I$ 0.873). In particular, D³M (AUROC$_I$ 0.890) uses a unique tri-modal data representation (combining 2D, Pseudo-3D and 3D inputs). In contrast, PIRN relies solely on two standard modalities: RGB images and surface normals. Despite utilizing fewer modalities, PIRN achieves a better localization performance and significantly outperforms D³M in specific categories, such as fork_crimp_terminal (0.978 vs. 0.819 AUROC$_I$) and lego_propeller (1.000 vs. 0.739 AUROC$_I$).

## 5 CONCLUSION

We introduced **PIRN**, a novel framework for few-shot multimodal anomaly detection that unifies prototype-based intra-modal reconstruction with cross-modal normality communication. **PIRN** models normality from scarce data via an adaptive prototype codebook. Its effectiveness comes from three key innovations: Balanced Prototype Assignment (BPA) utilizes balanced optimal transport to mitigate codebook collapse. Adaptive Prototype Refinement (APR) dynamically adapts prototypes during inference to bridge the train-test distribution gap. Multimodal Normality Communication (MNC) facilitates the exchange of high-level normality cues across modalities. Extensive evaluations across MVTec 3D-AD, Eyecandies, and Real-IAD D3 demonstrate that **PIRN** establishes significant performance gains in challenging few-shot settings.

## ACKNOWLEDGEMENTS

This work is supported by the Agency for Science, Technology and Research (A*STAR) under its MTC Programmatic Funds (Grant No. M23L7b0021).

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

## A  APPENDIX

Algorithm 1 provides pseudocode for the full **PIRN** pipeline.

---

**Algorithm 1** Pseudocode of the proposed **PIRN** framework.

---

1: **Input:** RGB image and surface-normal map
2: **Output:** Anomaly map (heatmap)
3: Extract multi-scale features $E_{rgb}$ and $E_{sn}$ using frozen ViT encoders
4: Initialize patch tokens: $Z_{rgb} \leftarrow E_{rgb}, \quad Z_{sn} \leftarrow E_{sn}$.
5: Randomly initialize prototypes $P_{rgb}$ and $P_{sn}$.
6: **for** each decoder layer $\ell = 1$ to $L$ **do**
7:     *Adaptive Prototype Refinement (APR)*
8:     Compute $\Gamma^*_{rgb}$ and $\Gamma^*_{sn}$ by solving the balanced OT in Eq. (1) for $(Z_{rgb}, P_{rgb})$ and $(Z_{sn}, P_{sn})$
9:     Compute context vectors $c^{rgb}_k$ and $c^{sn}_k$ for each prototype by column-normalized OT-weighted averaging of patch tokens
10:     Update each prototype $p^{rgb}_k$ and $p^{sn}_k$ using $\mathrm{GRU}(p_k, c_k)$
11:     *Balanced Prototype Assignment (BPA)*
12:     Compute $T^*_{rgb}$ and $T^*_{sn}$ by solving Eq. (1) for $(Z_{rgb}, P_{rgb})$ and $(Z_{sn}, P_{sn})$
13:     Reconstruct patch tokens: $Z^{bpa}_{rgb} = T^*_{rgb} P_{rgb}, \quad Z^{bpa}_{sn} = T^*_{sn} P_{sn}$     (Eq. (2))
14:     *MNC Stage 1: 2D and 3D Prototype Alignment*
15:     Align prototypes via cross-modal graph attention to obtain refined $P'_{rgb}$ and $P'_{sn}$
16:     *MNC Stage 2: Cross-Modal Feature Injection*
17:     Purify patch tokens: $Z'_{rgb} = Z_{rgb} \cdot \sigma(Z^{bpa}_{rgb}), \quad Z'_{sn} = Z_{sn} \cdot \sigma(Z^{bpa}_{sn})$
18:     Compute cross-attention outputs: $\mathrm{CA}(Z'_{rgb}, P'_{sn})$ and $\mathrm{CA}(Z'_{sn}, P'_{rgb})$     (Eq. (3))
19:     Apply gating to obtain $Z^{mnc}_{rgb}$ and $Z^{mnc}_{sn}$     (Eq. (4))
20:     Fuse reconstructions: $Z^{rec}_{rgb} = Z^{bpa}_{rgb} + Z^{mnc}_{rgb}, \quad Z^{rec}_{sn} = Z^{bpa}_{sn} + Z^{mnc}_{sn}$
21:     Update $Z_{rgb} \leftarrow Z^{rec}_{rgb}, \quad Z_{sn} \leftarrow Z^{rec}_{sn}$
22: **end for**
23: Compute per-modality patch anomaly scores: $d^{(rgb)}_i = 1 - \cos(E^{(rgb)}_i, Z^{rec}_{rgb,i})$ and $d^{(sn)}_i = 1 - \cos(E^{(sn)}_i, Z^{rec}_{sn,i})$
24: Fuse modalities to obtain the final anomaly map: $d_i = d^{(rgb)}_i + d^{(sn)}_i$

---

## B  MORE IMPLEMENTATION DETAILS

### B.1  DETAILS OF GATED PROTOTYPE UPDATE VIA GRU.

To ensure that prototypes coherently represent normal features, we update each prototype by incorporating this context vector $\{c_k\}$ through a gated recurrent unit (GRU) update. We treat the original prototype $p_k$ as the hidden state and its context $c_k$ as the input to a GRU cell, producing an updated prototype $p'_k$.

The GRU's gating mechanism dynamically controls the extent to which each prototype is updated, ensuring that only normal context is integrated while minimizing the incorporation of anomalous information during testing. Formally, the update for prototype $p_k$ is given by:

$$u_k = \sigma\Big( W_z \,[p_k; \; c_k] + b_z \Big), \tag{5}$$

$$r_k = \sigma\Big( W_r \,[p_k; \; c_k] + b_r \Big), \tag{6}$$

$$\tilde{p}_k = \tanh\Big( W \,[\, r_k \odot p_k; \; c_k] + b \Big), \tag{7}$$

$$p'_k = u_k \odot p_k \; + \; (1 - u_k) \odot \tilde{p}_k \,, \tag{8}$$

where $[\cdot; \; \cdot]$ denotes vector concatenation, $\odot$ is element-wise multiplication, $\sigma(\cdot)$ is the sigmoid activation, and $W_z, W_r, W$ (with corresponding biases $b_z, b_r, b$) are learnable weights. Eqs. (5)–(8)

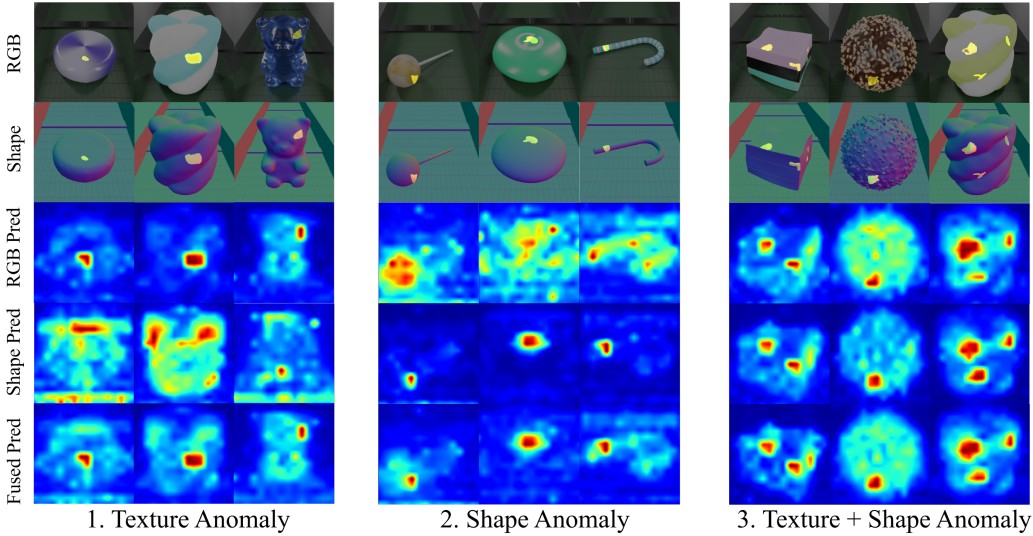

Figure 5: Qualitative results on Eyecandies with various types of anomalies, showing the complementary roles of the 2D and 3D branch in **PIRN**.

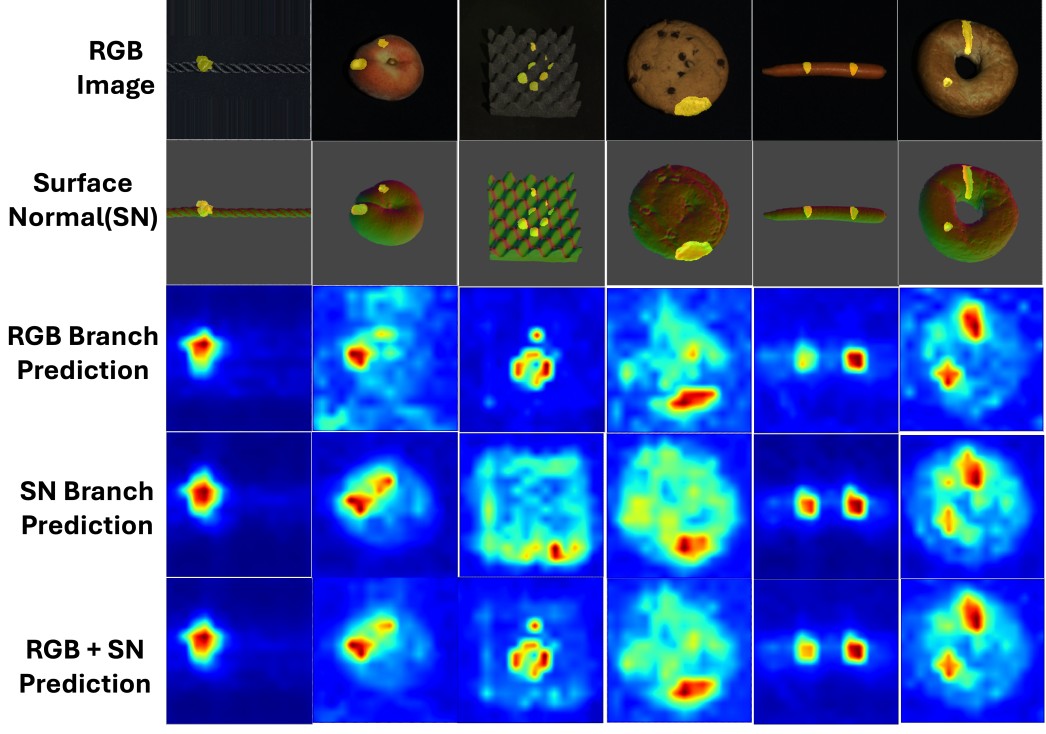

Figure 6: Localization performance on test samples with large or pervasive anomalies (10-shot normal training). PIRN accurately highlights extensive defects, demonstrating the APR module's robustness against prototype corruption even when anomalies dominate the input.

are the standard GRU equations adapted to our prototype refinement setting: $u_k$ is the update gate that decides how much of the previous prototype $p_k$ to keep, $r_k$ is the reset gate that modulates the influence of the past prototype when computing a candidate update, and $\tilde{p}_k$ is the candidate new

| Method | AUROC$_I$ | AUROC$_P$ | AUPRO |
|---|---|---|---|
| Softmax Attention | 0.832 | 0.967 | 0.929 |
| Linear Attention | 0.845 | 0.968 | 0.931 |
| Sigmoid Attention | 0.878 | 0.976 | 0.954 |
| Balanced Optimal Transport | 0.922 | 0.991 | 0.966 |

Table 9: Ablation of prototype assignment in BPA on **MVTec-3D-AD**.

| Backbone | AUROC$_I$ | AUROC$_P$ | AUPRO |
|---|---|---|---|
| DINOv1 ViT-B/8 | 0.892 | 0.974 | 0.946 |
| DINOv2 ViT-B/14 | 0.923 | 0.993 | 0.968 |
| DINOv2 ViT-L/14 | 0.928 | 0.994 | 0.970 |

Table 10: Comparison of anomaly detection and localization performance on **MVTec-3D-AD** under different backbones (10 shots).

prototype state. The final refined prototype $p'_k$ is a convex combination of the old prototype and the candidate state, weighted by the update gate.

This GRU-based update mechanism is crucial for maintaining robustness during prototype refinement. If the context $c_k$ aligns well with the original prototype, the GRU will produce a small $u_k$, allowing the new information $\tilde{p}_k$ to significantly override the old state $p_k$. Otherwise, if the context $c_k$ is unreliable due to an anomalous region that does not match any learned prototype, the update gate $u_k$ will be high (near 1) to keep the prototype unchanged. In this way, the GRU acts as a learnable gate: it adaptively suppresses anomalous information and only injects context when it is deemed normal.

### B.2 GENERATING SURFACE NORMAL IMAGES FROM POINT CLOUD.

We follow FIND's Li et al. (2025) procedure to generate surface normal maps. The MVTec 3D-AD dataset provides per-pixel 3D point cloud data for each sample. Using these organized point clouds, we compute a corresponding surface normal map. According to FIND Li et al. (2025), background pixels in point cloud are removed following the standard procedure in M3DM (Wang et al., 2023). For the foreground pixels, we use Open3D to compute normals by fitting a local plane to each point's neighborhood (KD-tree search with nearest neighbors $k = 30$). This produces an initial normal vector. We further enforce directional coherence using Open3D's `orient_normals _consistent_tangent_plane` with a connectivity setting of 50. After surface normal estimation, we project the computed normals back onto the image grid to form a normal map. For model training, we convert the normal vectors into a color image by linearly mapping each normal component to the $[0, 255]$ range. The resulting normal maps are saved as PNG images.

### B.3 HARDWARE FOR TRAINING.

All experiments were conducted on a workstation running Ubuntu 20.04 LTS. The system was equipped with two NVIDIA RTX 4090 GPUs with 128 GB of RAM. Our implementation was in Python 3.9 using PyTorch 1.13 (with CUDA 11.6 and cuDNN 8.1 for GPU acceleration). We also utilized OpenCV 4.8 for image processing and Open3D 0.17 for 3D data handling.

## C ABLATION STUDIES

### C.1 ADDITIONAL ABLATIONS ON BPA AND BACKBONE

We provide additional ablations analyzing (i) prototype assignment strategies in BPA and (ii) backbone architectures for the frozen encoders on **MVTec-3D-AD**. As shown in Tab. 9, softmax and linear attention yield the weakest results (AUROC$_I$ < 85%), suggesting that enforcing balanced prototype usage is crucial for stable reconstruction under limited data. As PIRN uses frozen encoders, feature quality impacts performance (Tab. 10). On 10-shot MVTec 3D-AD, DINOv2 (ViT-B/14) outperforms DINOv1 (ViT-B/8), improving AUROC$_I$ from 0.892 to 0.923 (+3.1%) due to richer semantic representations. Scaling to ViT-L/14 yields marginal gains (0.928). Crucially, PIRN remains robustly competitive even with the sub-optimal DINOv1.

### C.2 ROBUSTNESS TO LARGE-SCALE ANOMALIES.

Fig. 6 shows qualitative results on test samples where anomalies occupy a large portion of the object. PIRN remains robust because APR performs a constrained, single-step refinement around the learned

Table 11: Comparisons of per-category anomaly detection performance on MVTec-3D-AD.

| Method | Bagel | Cable Gland | Carrot | Cookie | Dowel | Foam | Peach | Potato | Rope | Tire | Mean |
|---|---|---|---|---|---|---|---|---|---|---|---|
| **AUROC$_I$** | | | | | | | | | | | |
| BTF (Horwitz & Hoshen, 2023) | 0.938 | 0.765 | 0.972 | 0.888 | 0.960 | 0.664 | 0.904 | 0.929 | 0.982 | 0.726 | 0.865 |
| AST (Rudolph et al., 2023) | 0.983 | 0.873 | 0.976 | 0.971 | 0.932 | 0.885 | 0.974 | **0.981** | **1.000** | 0.797 | 0.937 |
| M3DM (Wang et al., 2023) | 0.994 | 0.909 | 0.972 | 0.976 | 0.960 | 0.942 | 0.973 | 0.899 | 0.972 | 0.850 | 0.945 |
| CFM (Costanzino et al., 2024) | 0.994 | 0.888 | **0.984** | **0.993** | **0.980** | 0.888 | 0.941 | 0.943 | 0.980 | **0.953** | 0.954 |
| 3D-ADNAS (Long et al., 2025) | **0.997** | **1.000** | 0.971 | 0.986 | 0.966 | 0.948 | 0.897 | 0.873 | **1.000** | 0.867 | 0.951 |
| Shape Guided (Chu et al., 2023) | 0.986 | 0.894 | 0.983 | 0.991 | 0.976 | 0.857 | 0.990 | 0.965 | 0.990 | 0.869 | 0.947 |
| **PIRN** | 0.971 | 0.973 | 0.941 | 0.957 | 0.975 | **0.993** | **0.992** | 0.950 | 0.996 | 0.880 | **0.963** |
| **AUPRO@30%** | | | | | | | | | | | |
| BTF (Horwitz & Hoshen, 2023) | 0.976 | 0.969 | 0.979 | **0.973** | 0.933 | 0.888 | 0.896 | 0.912 | 0.950 | 0.971 | 0.959 |
| AST (Rudolph et al., 2023) | 0.970 | 0.947 | 0.981 | 0.939 | 0.913 | 0.906 | 0.979 | 0.982 | 0.889 | 0.940 | 0.944 |
| M3DM (Wang et al., 2023) | 0.970 | 0.971 | 0.979 | 0.950 | 0.941 | 0.932 | 0.977 | 0.971 | 0.971 | 0.975 | 0.964 |
| CFM (Costanzino et al., 2024) | 0.979 | 0.972 | 0.982 | 0.945 | 0.950 | 0.968 | 0.980 | 0.943 | 0.950 | **0.981** | 0.971 |
| Shape Guided (Chu et al., 2023) | **0.981** | 0.973 | 0.982 | 0.971 | 0.962 | **0.978** | 0.981 | **0.983** | 0.974 | 0.975 | **0.976** |
| **PIRN** | 0.966 | **0.978** | **0.983** | 0.972 | **0.976** | 0.971 | **0.981** | 0.978 | **0.974** | 0.951 | 0.973 |

normal codebook rather than unconstrained test-time re-learning. First, APR extracts prototype contexts via entropy-regularized optimal transport, which assigns negligible mass to patches that are dissimilar to all normal prototypes, thereby limiting the influence of anomalous regions. Second, the GRU gate (trained only on normal data) tends to close when the context is unreliable, preventing prototype corruption.

### C.3 PER-CATEGORY RESULTS ON MVTEC-3D-AD.

PIRN demonstrates strong detection performance on the MVTec-3D-AD dataset (Table 11). It achieves a mean AUROC$_I$ of **0.963**, surpassing the strongest baseline CFM (0.954) and the previous state-of-the-art 3D-ADNAS (0.951). It also outperforms other representative baselines such as M3DM (0.945) and AST (0.937). These results demonstrate that PIRN not only improves average detection accuracy but also localizes diverse texture and shape defects in complex anomaly detection scenarios.

## D VISUALIZATION OF ANOMALY LOCALIZATION.

Fig. 7 and Fig. 8 illustrate PIRN's anomaly localization results on the Eyecandies and MVTec 3D-AD datasets, respectively. These qualitative results highlight PIRN's robustness in capturing fine-grained defect details while avoiding false positives.

## E LLM USAGE DISCLOSURE.

We used a large language model to support language editing for the camera ready version, including improving clarity, grammar, and overall presentation of the manuscript. All scientific contributions, such as research conception, methodological design, experimental setup, result analysis, and interpretation—were carried out by the authors, who take full responsibility for the content.

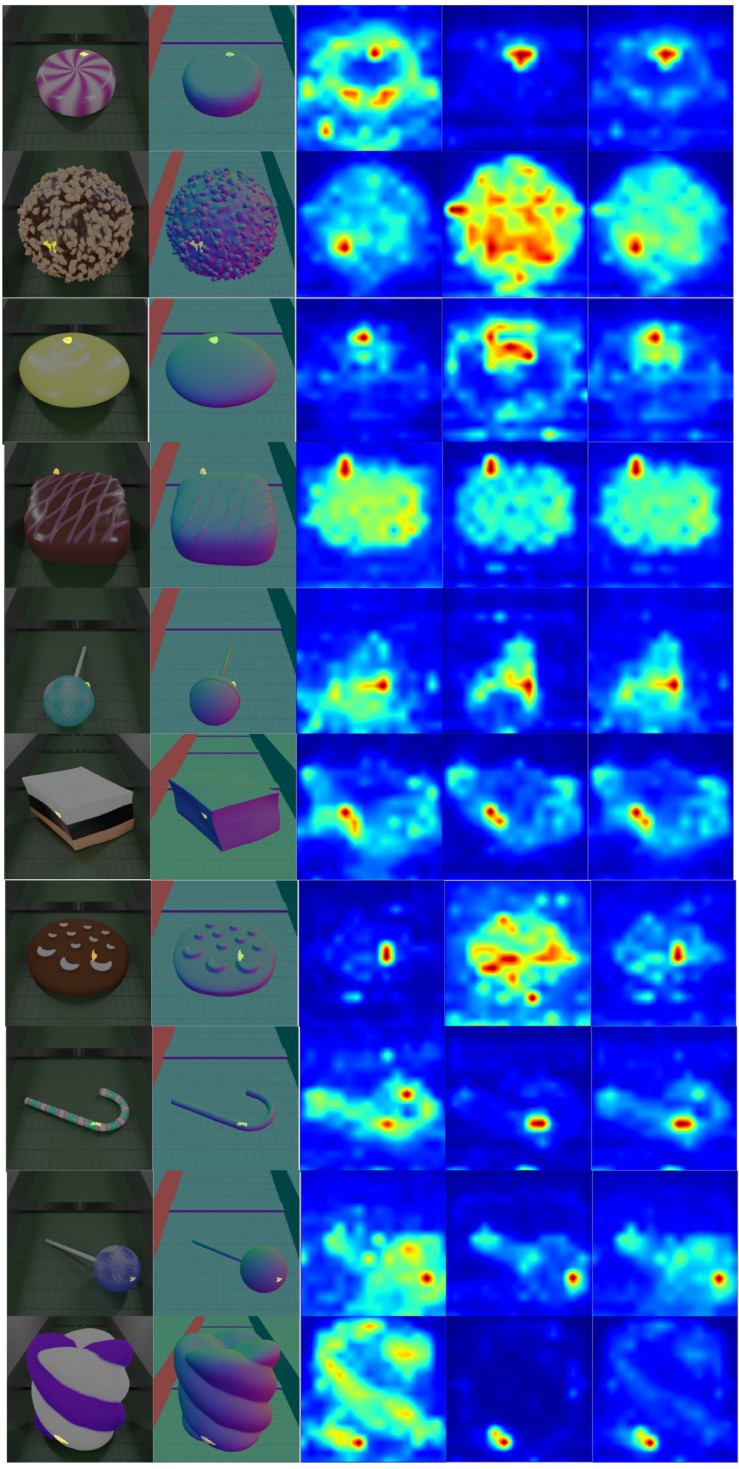

Figure 7: Visualization of localization performance on the Eyecandies dataset (5-shot normal training). From left to right: RGB images, surface normals, anomaly maps predicted by the RGB branch, anomaly maps predicted by the surface-normal branch, the fused anomaly map.

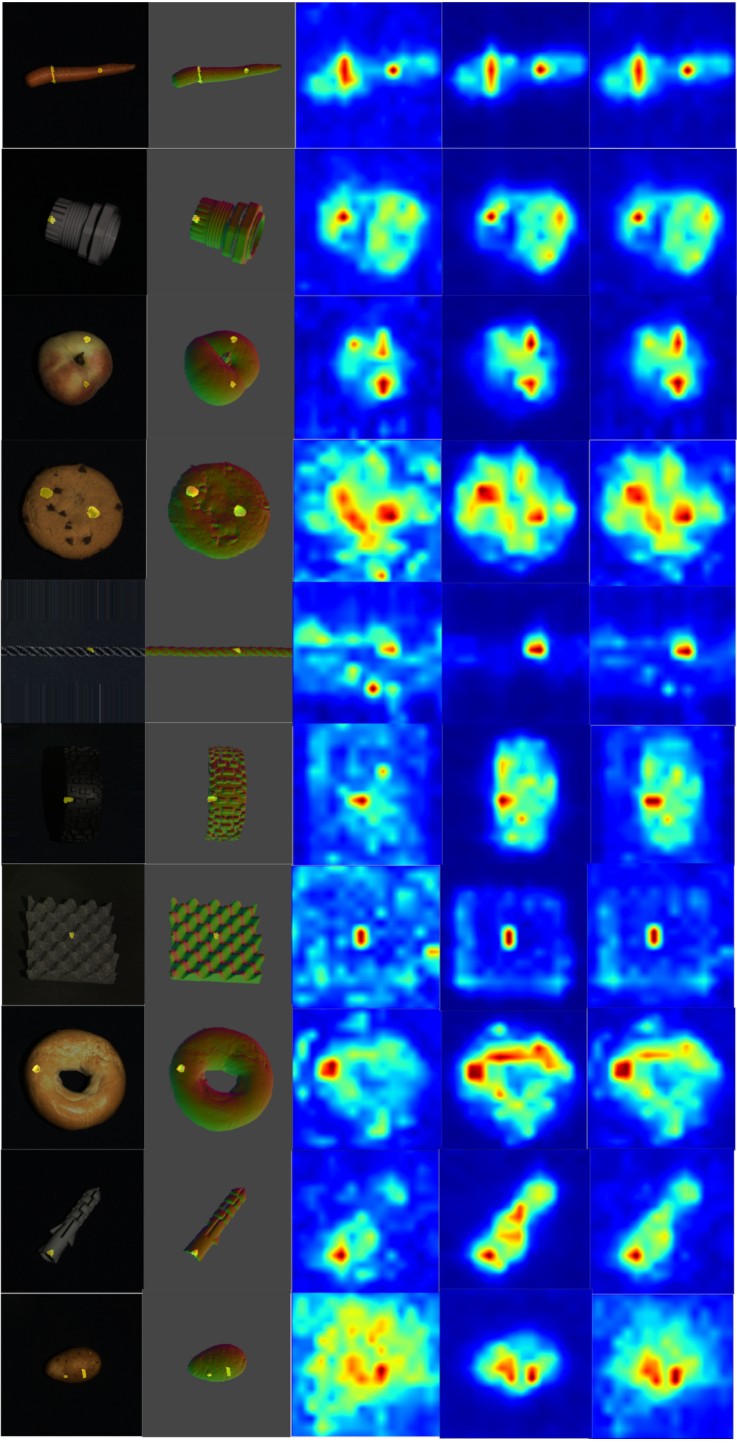

Figure 8: Visualization of localization performance on the MVTec 3D-AD dataset (5-shot normal training). From left to right: RGB images, surface normals, anomaly maps predicted by the RGB branch, anomaly maps predicted by the surface-normal branch, the fused anomaly map.

