# OpenReview forum: "PIRN: Prototypical-based Intra-modal Reconstruction with Normality Communication for Multi-modal Anomaly Detection."
_ICLR.cc/2026/Conference — ICLR 2026 Poster_

### Official Review · Reviewer_tn1N · 2025-10-17

**Soundness:** 2
**Presentation:** 3
**Contribution:** 2
**Rating:** 4
**Confidence:** 4

**Summary:**

This paper introduces PIRN, a framework for few-shot multi-modal anomaly detection that leverages a prototype-based intra-modal reconstruction approach. The core contribution lies in three synergistic components: Balanced Prototype Assignment (BPA) to mitigate codebook collapse, Adaptive Prototype Refinement (APR) to handle unseen normal variations at test time, and Multi-modal Normality Communication (MNC) to enable knowledge transfer between modalities. By focusing on reconstructing features from a compact, adaptive codebook of normal patterns, PIRN addresses the limitations of existing methods in data-scarce scenarios and achieves new state-of-the-art performance on benchmark datasets.

**Strengths:**

- The paper addresses a practical problem of few-shot multi-modal anomaly detection in industrial quality control. The proposed prototype-based reconstruction paradigm is an elegant solution.
- The technical contributions are clearly presented. Each of the three key components (i.e., BPA, APR, MNC) is designed to solve a specific challenge.
- The experimental evaluation demonstrates significant performance gains over strong baselines across multiple datasets and various few-shot settings. The ablation studies effectively validate the contribution of each proposed module.

**Weaknesses:**

- My main concerns lie in the potential computational complexity and practical deployability of the proposed PIRN framework. The model incorporates multiple computationally intensive steps within each decoder layer, including two iterative optimal transport calculations for BPA and APR, and a graph attention network for MNC. The paper lacks any discussion on inference latency or FLOPs, which is a critical metric for real-world industrial inspection systems that often require high throughput. This omission makes it difficult to assess whether the impressive accuracy comes at the cost of practical usability.
- The robustness of the APR module relies on the assumption that anomalies are sparse. Its effectiveness might degrade in cases of large-scale anomalies, where the *normal context* itself becomes contaminated.
- The performance of the model seems highly dependent on several key hyperparameters, most notably the number of prototypes *K*. No sensitivity analysis is provided to show how performance varies with these choices.

**Questions:**

- Have you investigated the failure cases of the APR module when dealing with test samples containing large or pervasive anomalies? Is there a mechanism to prevent prototype corruption in such scenarios?
- BPA is designed to prevent prototype under-utilization, so I am wondering that is there a risk of learning *redundant* prototypes when *K* is large relative to the few-shot training data, where multiple prototypes capture nearly identical normal patterns?
- How much does the performance of PIRN depend on the quality of this pre-training (e.g., DINOv2 vs. standard ImageNet pre-training)?

---

> ### Author Response · Authors · 2025-12-01
>
> **My main concerns lie in the potential computational complexity and practical deployability of the proposed PIRN framework. The model incorporates multiple computationally intensive steps within each decoder layer, including two iterative optimal transport calculations for BPA and APR, and a graph attention network for MNC. The paper lacks any discussion on inference latency or FLOPs, which is a critical metric for real-world industrial inspection systems that often require high throughput. This omission makes it difficult to assess whether the impressive accuracy comes at the cost of practical usability.**
>
> | **Method**         | **AUROC\_I↑** | **FLOPs↓ (G)** | **Latency↓ (ms)** | **Peak Memory↓ (GB)** |
> |--------------------|--------------|----------------|-------------------|------------------------|
> | M3DM [1](2023)        | 0.845        | 263.27         | 21.61             | **1.26**               |
> | CFM [2](2024)         | 0.845        | 552.53         | 60.57             | 2.35                   |
> | FIND [3](SOTA)        | 0.921        | 728.46         | 76.09             | 3.28                   |
> | **PIRN (Ours)**    | **0.922**    | **103.36**     | **17.49**         | 1.28                   |
>
> We thank the reviewer for raising the concern regarding model complexity. To clarify the trade-off between complexity and performance, we conducted a detailed efficiency comparison against recent reconstruction-based MAD baselines (M3DM[1], CFM[2], and the recent SOTA method FIND[3]) on the 10-shot MVTec-3D-AD setting. We evaluated accuracy (AUROC$_I$) and efficiency (FLOPs, latency, and Peak GPU Memory) with input $224\times224$ and batch size 1. As summarized in the above Table, PIRN achieves the highest AUROC$_I$ (0.922) while simultaneously being the most efficient method across all metrics. PIRN requires only 103.36G FLOPs, significantly less than M3DM (263.27G), CFM (552.53G), and FIND (728.46G).
>
> Contrary to the initial impression of complexity, our architectural design choices lead to substantial efficiency gains. Specifically, PIRN achieves SOTA accuracy with **85\% fewer FLOPs** and **4.5x faster latency** (17.49ms vs 76.09ms) compared to FIND, alongside the lowest memory footprint (1.09GB). This efficiency stems from two key factors. First, unlike methods (CFM, M3DM) that use expensive PointNet-style backbones on raw point clouds, PIRN utilizes efficient 3D encoding by converting point clouds into surface-normal images and applying a 2D ViT (DINO V2). Second, PIRN employs a lightweight decoder. While CFM uses heavy convolutional upsampling and FIND relies on dense token-to-token distillation blocks (728.46G FLOPs), PIRN avoids these heavy operations. Our core modules (BPA, APR, MNC) operate primarily on a compact set of prototypes ($K=10$) rather than all patch tokens ($N=256$). The complexity of these operations (e.g., OT routing, graph attention) is $O(NK + K^2)$, which is negligible compared to the backbone.
>
>
>
> [1] Wang, Y., Peng, J., Zhang, J., Yi, R., Wang, Y. and Wang, C., 2023. Multimodal industrial anomaly detection via hybrid fusion. CVPR
>
> [2] Costanzino, A., Ramirez, P.Z., Lisanti, G. and Di Stefano, L., 2024. Multimodal industrial anomaly detection by crossmodal feature mapping.CVPR
>
> [3] Li, Y., Liu, F., Liao, J., Tian, S., Foo, C.S. and Yang, X., 2025. FIND: Few-Shot Anomaly Inspection with Normal-Only Multi-Modal Data. ICCV
>
> **The performance of the model seems highly dependent on several key hyperparameters, most notably the number of prototypes K. No sensitivity analysis is provided to show how performance varies with these choices.**
>
> | **Number of Prototypes** | **AUROC\_I** | **AUROC\_P** | **AUPRO** | **F1max\_I** |
> |--------------------------|--------------|--------------|-----------|--------------|
> | K = 5                    | 0.954        | 0.993        | 0.969     | 0.957        |
> | **K = 10**               | **0.964**    | **0.994**    | **0.973** | **0.962**    |
> | K = 50                   | 0.940        | 0.991        | 0.966     | 0.952        |
> | K = 100                  | 0.901        | 0.984        | 0.950     | 0.947        |
>
>
> The results indicate that model performance is sensitive to the choice of $K$. When $K$ is too small (e.g., $K=5$), the codebook capacity appears insufficient to represent the full distribution of normal variations, leading to suboptimal performance (0.954 I-AUROC).Conversely, significantly increasing the number of prototypes (e.g., $K=50$ or $K=100$) leads to a noticeable performance drop (I-AUROC decreasing to 0.940 and 0.901, respectively). This suggests that an excessively large codebook may lead to overfitting or inefficient utilization, potentially capturing spurious details rather than essential normal patterns, which hinders anomaly discrimination.The best performance across all metrics (0.964 I-AUROC, 0.973 AUPRO) is achieved when $K=10$. Therefore, we adopted $K=10$ as the default setting for our experiments.

---

> > ### Author Response · Authors · 2025-12-01
> >
> > **Q:Have you investigated the failure cases of the APR module when dealing with test samples containing large or pervasive anomalies? Is there a mechanism to prevent prototype corruption in such scenarios?**
> >
> >
> >
> > It is crucial to clarify that the Adaptive Prototype Refinement (APR) module is not designed to re-learn prototypes during testing. Instead, its objective is to perform a localized, single-step adaptation around the established normality codebook learned during training. The robustness of APR against large anomalies comes from three constraints in our design.
> >
> > **1. Inherent OOD Suppression by Optimal Transport (OT).**
> > The context extraction mechanism in APR is intrinsically resistant to Out-of-Distribution (OOD) tokens. Before updating a prototype $p_k$, APR computes a context vector $c_k$ by solving an entropy-regularized OT problem (Eq. 3) between the input patch tokens and the prototypes. Anomalous patches that are highly dissimilar to all learned normal prototypes will incur high transport costs. Consequently, they receive negligible transport mass (weights) in the resulting OT plan. Furthermore, the OT formulation employs uniform marginal constraints (Eq. 1), ensuring that each prototype receives a balanced share of the total mass and preventing the concentration of mass onto a few prototypes. As a result, even when large anomalous regions are present, their influence on the context vectors (Eq. 3) is minimal and diluted, resembling uniform noise. The update direction remains dominated by the ``normal context'' formed by in-distribution patches.
> >
> > **2. Conservative GRU Gating Trained on Normal Data.**
> > The refinement process utilizes a Gated Recurrent Unit (GRU), where the prototype $p_k$ acts as the hidden state and the context vector $c_k$ as the input. Crucially, the GRU is trained only on normal samples; it has only observed scenarios where ``normal context leads to reasonable prototype updates.'' During testing, if an image contains pervasive anomalies, the corresponding context vector will likely fall outside the distribution encountered during training. When faced with such unfamiliar input, the GRU's update gate tends to remain closed. As detailed in Section 3.3 (Lines 281-285), this mechanism ensures that if the context is unreliable, the gate remains largely closed, leaving the prototype essentially unchanged. This conservative gating allows adaptation to unseen normal variations while preventing drastic updates toward anomalous directions.
> >
> >
> >
> > **3. Constraint via Information Bottleneck.**
> > APR operates within the broader PIRN framework, where the prototypes constitute a compact codebook of normality, learned via the Balanced Prototype Assignment (BPA) bottleneck. This defines a constrained ``normality subspace.'' APR merely facilitates minor adjustments within this subspace; it is not unconstrained optimization. Even if a pervasive anomaly causes a slight shift in the prototypes, the subsequent reconstruction step (BPA, Section 3.2) forces all features to be projected onto this discrete codebook. Anomalies fundamentally cannot be faithfully reconstructed through this strong information bottleneck. Therefore, the architecture—constrained updates (APR) followed by projection onto the normal codebook (BPA)—ensures that anomalies will still manifest as large reconstruction errors.
> >
> >
> > Finally, we visualized the localization results of large anomalies under the 10-shot setting and attached as Fig 6 in appendix. The visualization includes challenging cases where anomalies occupy a substantial portion of the input. As shown in Fig,  the resulting anomaly maps remain highly precise while minimizing false positives in the normal background. This indicates that the prototypes are not significantly affected by the large anomalies during adaptation. This also validates that APR facilitates localized adaptation rather than unrestricted re-learning, ensuring excellent performance even when processing inputs dominated by large-scale defects.

---

> ### Author Response · Authors · 2025-12-01
>
> **Q: BPA is designed to prevent prototype under-utilization, so I am wondering that is there a risk of learning redundant prototypes when K is large relative to the few-shot training data, where multiple prototypes capture nearly identical normal patterns?**
>
> While increasing the codebook size $K$ in standard prototype-based methods can indeed lead to redundancy, especially in few-shot scenarios, PIRN incorporates several mechanisms that can encourage prototype diversity and specialization to mitigate this risk.
>
>
>
> **1. BPA Enforces Equal-Mass Clustering and Encourages Divergence.**
> By formulating patch-to-prototype matching as a balanced Optimal Transport (OT) problem with strict uniform utilization constraints (each prototype must receive $\approx 1/K$ of the total mass), BPA effectively transforms the learning process into an ``equal-mass clustering'' problem. This prevents prototypes from collapsing onto the same high-density regions of the normal manifold. If two prototypes were to become nearly identical, their receptive fields would overlap completely, leading to unstable mass allocation in the OT solution. The optimization process, driven by minimizing both the reconstruction loss and the OT cost, will push these prototypes apart so that each uniquely captures a distinct subset of patches. As visualized in Fig. 1 (Right), BPA yields prototypes uniformly distributed across the normal manifold, contrasting sharply with the clustered prototypes observed under softmax assignment.
>
> **2. APR Drives Fine-Grained Specialization.**
> The Adaptive Prototype Refinement (APR, Section 3.3) further drives specialization. Even if two prototypes are initially similar, the balanced OT constraint ensures they are assigned slightly different subsets of patches, resulting in distinct context vectors $c_k$ (Eq. 3). The iterative, per-prototype GRU updates amplify these subtle differences over time. This dynamic process pulls the prototypes toward distinct, localized normal modes (e.g., variations in texture scale, position, or lighting), promoting a fine-grained coverage of the normal distribution rather than redundancy.
>
>
> To empirically validate the optimal codebook size and investigate the impact of $K$, we conducted an ablation study on the MVTec 3D-AD dataset (all-shot setting), summarized in the above table. The results demonstrate a clear trade-off between representational capacity and the strength of the information bottleneck. Increasing $K$ from 5 to 10 improves the image-level AUROC from 0.954 to 0.964, indicating that $K=5$ provides insufficient capacity to capture the full diversity of normal patterns. However, performance significantly degrades when $K$ is increased further to 50 (0.940) and 100 (0.901). This degradation confirms the importance of maintaining a compact codebook. An excessively large $K$ weakens the crucial information bottleneck effect, expanding the model's capacity such that anomalies may be partially reconstructed (the ``identity shortcut''), thereby reducing the discriminative power of the anomaly scores. Based on these findings, we selected $K=10$ as the optimal balance.
>
> **Q:How much does the performance of PIRN depend on the quality of this pre-training, e.g., DINOv2 vs. standard ImageNet pre-training**
>
> | **Backbone**                    | **AUROC\_I** | **AUROC\_P** | **AUPRO** | **F1max\_I** |
> |---------------------------------|--------------|--------------|-----------|--------------|
> | DINOv1 ViT-B/8                  | 0.892        | 0.974        | 0.946     | 0.923        |
> | **DINOv2 ViT-B/14 (default)**   | 0.923        | 0.993        | 0.968     | 0.951        |
> | DINOv2 ViT-L/14                 | 0.928        | 0.994        | 0.970     | 0.952        |
>
>
> As PIRN utilizes frozen encoders, the quality of the foundational features indeed influences performance. To quantify this dependency, we conducted an ablation study on the MVTec 3D-AD dataset (10-shot setting). Utilizing the more advanced DINOv2 (ViT-B/14) substantially outperforms DINOv1 (ViT-B/8, specifically the "vit\_base\_patch8\_224.dino" checkpoint trained on ImageNet-1k), improving the AUROC$_I$ from 0.892 to 0.923 (+3.1\%). This highlights that the richer semantic and geometric representations captured by advanced self-supervised training enhance PIRN's capabilities.
>
> However, when examining the impact of model scale within the DINOv2 family, the gains saturate rapidly. Scaling up from ViT-B to ViT-L yields only a marginal improvement (0.923 to 0.928). Crucially, even when using the less optimal DINOv1 backbone, PIRN still achieves a competitive AUROC$_I$ of 0.892. This performance surpasses the strongest baseline reported in the 10-shot setting (INP-Former, 0.885, Table 1). Therefore, while high-quality pre-training is essential for achieving state-of-the-art results, the PIRN framework demonstrates robustness and effectiveness across different backbone architectures.

---

### Official Review · Reviewer_sM1V · 2025-10-26

**Soundness:** 3
**Presentation:** 3
**Contribution:** 3
**Rating:** 6
**Confidence:** 4

**Summary:**

The paper proposes a multimodal anomaly detection method (RGB + Surface normals) named PIRN. The method is based on the idea that every image can be reconstructed without anomalies with correctly chosen “prototypes” if it is only trained with normal data. The paper proposes two main ways to improve the baseline method (INP-Former): the first one lies in the improvement of the extracted prototypes (balancing + refinement), and the other lies in combining the prototypes extracted from the RGB image and the surface normals. The method is then evaluated on two popular datasets, MVTec3D and Eyecandies, in both low-shot (the authors use the word few-shot, but that is typically reserved for k <= 8) and full-data regimes. In both regimes, they achieve great results

**Strengths:**

- Cross-modal prototype interaction is a well-thought-out solution. The results in the ablation study confirm this
- The model works extremely well in a low-shot scenario.

**Weaknesses:**

- The method fails to compare with some methods that outperform it in the full-shot scenario on MVTec 3D: 3DSR (WACV 24) and TransFusion (ECCV 2024). As these two methods are trained with synthetic anomalies, I would assume they do not work as good as PIRN in the low-shot scenario. I would recommend adding them at least to Table 5.
- It is unclear how INP-Former is trained for comparison on MVTec 3D. Are both modalities input, or only one?
- The number of prototypes is not specified anywhere
- It is unclear how the model performs in scenarios where only RGB or only Surface Normals are available
- How robust is the model to the choice of the feature extractor? Let us say, what if we used a DINOv3 ViT instead of the “standard” ViT?
- Computational complexity is not compared to other methods

**Questions:**

I have several questions. I have sorted them from most problematic to least problematic.

1. What is the performance of PIRN in scenarios where only RGB or only Surface Normals are available?
2. Does INP-Former get only the RGB as the input or also the surface normals? Is the performance improved if the surface normals are used as input as well?
3. How much time is required to train PIRN, and what is the inference speed of PIRN? What is the inference speed of the model without APR?
4. How many prototypes are used?
5. How does PIRN work with other backbones?

---

> ### Author Response · Authors · 2025-12-01
>
> **Q: What is the performance of PIRN in scenarios where only RGB or only Surface Normals are available?**
>
> | **Setting / Modality**          | **AUROC\_I** | **AUROC\_P** | **AUPRO** | **F1max\_I** |
> |---------------------------------|--------------|--------------|-----------|--------------|
> | **5-shot**                      |              |              |           |              |
> | RGB-only                        | 0.794        | 0.966        | 0.890     | 0.910        |
> | Surface Normals-only            | 0.854        | 0.972        | 0.912     | 0.932        |
> | **RGB + Surface Normals**       | 0.894        | 0.992        | 0.962     | 0.940        |
> | **10-shot**                     |              |              |           |              |
> | RGB-only                        | 0.827        | 0.968        | 0.901     | 0.921        |
> | Surface Normals-only            | 0.879        | 0.974        | 0.920     | 0.937        |
> | **RGB + Surface Normals**       | 0.923        | 0.993        | 0.968     | 0.951        |
> | **All-shot**                    |              |              |           |              |
> | RGB-only                        | 0.874        | 0.977        | 0.917     | 0.929        |
> | Surface Normals-only            | 0.937        | 0.977        | 0.928     | 0.946        |
> | **RGB + Surface Normals**       | 0.964        | 0.994        | 0.973     | 0.962        |
>
>
>
> To evaluate PIRN's performance when only one modality is available, we conducted additional experiments on MVTec-3D-AD. We report the results under 5-shot, 10-shot, and all-shot settings for RGB-only, Surface Normals (SN)-only, and the combined RGB+SN configurations in the above table.
>
>
> Single-Modality Performance. The results demonstrate that PIRN maintains considerable performance even when restricted to a single modality. For instance, in the 10-shot setting, the I-AUROC is 0.827 (RGB-only) and 0.879 (SN-only). Notably, SN-only consistently outperforms RGB-only across all settings (by approximately 5–6\% I-AUROC). This aligns with the observation that 3D geometric information is often more sensitive to the structural defects prevalent in industrial inspection.
>
> Multi-Modal Fusion Benefits. Crucially, when both RGB and Surface Normals are utilized, PIRN significantly surpasses the performance of either single modality. The integration of multi-modal information yields substantial improvements in I-AUROC over the best single modality (SN-only). In conclusion, PIRN effectively leverages the complementary nature of RGB and Surface Normals, leading to significant and consistent performance gains in both few-shot and all-shot scenarios.
>
> **Q: Does INP-Former get only the RGB as the input or also the surface normals? Is the performance improved if the surface normals are used as input as well?**
>
> We clarify that the INP-Former results reported in Table 1 are based on a multi-modal implementation (RGB + Surface Normals), not RGB-only. To ensure a fair comparison for the MAD task, we adapted the official 2D INP-Former (Luo et al., 2025a) into a two-stream architecture. This baseline utilizes two independent branches, one for RGB and one for surface-normal maps. Crucially, both branches employ the exact same configuration as PIRN, including the frozen DINOv2 ViT-Base/14 backbone, input resolution ($224\times224$), number of decoder layers and $k$-shot training splits. During inference, each branch independently generates an anomaly map. These maps are then fused via element-wise summation to produce the final result. This entire inference, fusion, and post-processing pipeline is identical to the procedure used for PIRN (Sec. 3.4).Therefore, both PIRN and the multi-modal INP-Former baseline operate on the same input information and use the same scoring protocol. The significant performance advantage demonstrated by PIRN (Table 1) stems directly from our architectural innovations—namely the Balanced Prototype Assignment (BPA), Adaptive Prototype Refinement (APR), and explicit Multimodal Normality Communication (MNC)—rather than the utilization of different modalities or evaluation strategies. We will revise the manuscript to detail this multi-modal baseline setup explicitly in the experimental section.

---

> > ### Author Response · Authors · 2025-12-01
> >
> > **Q: How much time is required to train PIRN, and what is the inference speed of PIRN? What is the inference speed of the model without APR?**
> >
> > | **Setting / Method**       | **Train Time↓ (h)** | **F1max\_I** | **Inference FLOPs↓ (G)** | **Inference Latency↓ (ms)** | **Peak Memory↓ (ms)** |
> > |----------------------------|---------------------|-------------|---------------------------|-----------------------------|------------------------|
> > | 5-shot PIRN                | 0.040               | 0.940       | 103.055                   | 18.519                      | 1.275                  |
> > | 10-shot PIRN               | 0.062               | 0.951       | 103.055                   | 17.087                      | 1.275                  |
> > | All-shot PIRN              | 1.560               | 0.964       | 111.131                   | 35.625                      | 1.991                  |
> > | 5-shot PIRN w/o APR        | 0.032               | 0.933       | 102.740                   | 16.231                      | 0.00                   |
> > | 10-shot PIRN w/o APR       | 0.053               | 0.948       | 102.740                   | 15.802                      | 1.275                  |
> > | All-shot PIRN w/o APR      | 0.925               | 0.962       | 110.188                   | 27.560                      | 1.997                  |
> >
> >
> > We have summarized the training time and inference efficiency in the above table. PIRN is highly efficient to train, particularly in few-shot scenarios. The 5-shot and 10-shot configurations require only approximately 0.040h and 0.062h , respectively. The all-shot setting takes about 1.56 hours for  the full training set. In the 10-shot setting, the model processes an input pair in 17.087ms (approx. 58.5 FPS) with 103.055 GFLOPs. The all-shot model, which utilizes a deeper architecture (8 decoder layers), requires 35.625ms (approx. 28.1 FPS). To evaluate the overhead of the APR module, we analyzed the model without it. Removing APR reduces the 10-shot latency to 15.80 ms. This indicates that using APR introduces a small overhead ($\sim$1.3ms) while improving the F1max$_I$ score from 0.948 to 0.951. Given the performance improvements and negligible impact on inference speed, APR is an efficient component for enhancing model adaptability.
> >
> > **Q: How many prototypes are used?**
> >
> > | **Number of Prototypes** | **AUROC\_I** | **AUROC\_P** | **AUPRO** | **F1max\_I** |
> > |--------------------------|--------------|--------------|-----------|--------------|
> > | K = 5                    | 0.954        | 0.993        | 0.969     | 0.957        |
> > | **K = 10**               | **0.964**    | **0.994**    | **0.973** | **0.962**    |
> > | K = 50                   | 0.940        | 0.991        | 0.966     | 0.952        |
> > | K = 100                  | 0.901        | 0.984        | 0.950     | 0.947        |
> >
> >
> > The results indicate that model performance is sensitive to the choice of $K$. When $K$ is too small (e.g., $K=5$), the codebook capacity appears insufficient to represent the full distribution of normal variations, leading to suboptimal performance (0.954 I-AUROC).Conversely, significantly increasing the number of prototypes (e.g., $K=50$ or $K=100$) leads to a noticeable performance drop (I-AUROC decreasing to 0.940 and 0.901, respectively). This suggests that an excessively large codebook may lead to overfitting or inefficient prototype utilization.The best performance across all metrics (0.964 I-AUROC, 0.973 AUPRO) is achieved when $K=10$. Therefore, we adopted $K=10$ as the default setting for our experiments.
> >
> > **Q: How does PIRN work with other backbones?**
> >
> > | **Backbone**                    | **AUROC\_I** | **AUROC\_P** | **AUPRO** | **F1max\_I** |
> > |---------------------------------|--------------|--------------|-----------|--------------|
> > | DINOv1 ViT-B/8                  | 0.892        | 0.974        | 0.946     | 0.923        |
> > | **DINOv2 ViT-B/14 (default)**   | 0.923        | 0.993        | 0.968     | 0.951        |
> > | DINOv2 ViT-L/14                 | 0.928        | 0.994        | 0.970     | 0.952        |
> >
> > We evaluated PIRN using various ViT architectures in the 10-shot setting on MVTec 3D-AD. For a fair comparison, our default backbone (DINOv2 ViT-B/14) aligns with the configuration used in the baseline INP-Former. When switching to DINOv1 ViT-B/8 (the $vit\_base\_patch8\_224.dino$ model pretrained on ImageNet-1k), performance decreases (89.2\% vs. 92.3\% I-AUROC). This indicates that PIRN benefits significantly from the more generalized features provided by DINOv2. Notably, even with the DINOv1 backbone, PIRN (89.2\%) still outperforms the other baselines (e.g., CFM, 84.5\% I-AUROC).
> >
> > Furthermore, employing a deeper model DINOv2 ViT-L/14 yields a marginal improvement (92.8\% I-AUROC). These results demonstrate that while the quality of the pretrained features impacts overall performance, the PIRN framework is robust and generalizes well across different ViT architectures.

---

### Official Review · Reviewer_HJaY · 2025-10-29

**Soundness:** 3
**Presentation:** 2
**Contribution:** 2
**Rating:** 6
**Confidence:** 4

**Summary:**

Existing multimodal anomaly detection methods often perform poorly in few-shot scenarios. To address this issue, the paper introduces a prototype-driven, reconstruction-based method. It leverages prototypes to build a codebook that filters out anomalous features in the bottleneck. The method also aligns RGB and depth information, integrating multimodal data into the reconstruction process. Finally, anomalies are localized based on the reconstruction error.

**Strengths:**

+ The proposed codebook-based reconstruction approach is well-suited for the few-shot problem. Additionally, the optimal transport strategy makes excellent use of the limited available normal samples.
+ The idea of having modalities mutually assist in reconstruction is highly novel, and the results show it to be quite effective.
+ The experiments in both few-shot and all-shot settings effectively demonstrate the method's validity. The ablation study is also thorough and convincing.

**Weaknesses:**

+ The prototype-based reconstruction method is quite similar to the approaches in some previous works [1, 2]. However, these papers were not cited.
+ The proposed design appears to be quite complex. Its computational complexity seems significantly higher than that of previous reconstruction-based methods. It would be helpful to know if the authors have considered this trade-off.
+ The paper lacks a clear justification for the choice of the number of decoder layers (2 for few-shot and 8 for all-shot). The authors should consider adding this as a factor in the ablation study.
+ There is a minor issue of a duplicate citation for the paper "Revisiting multimodal fusion for 3d anomaly detection from an architectural perspective."

[1] Gong, D., Liu, L., Le, V., Saha, B., Mansour, M. R., Venkatesh, S., & Hengel, A. V. D. (2019). Memorizing normality to detect anomaly: Memory-augmented deep autoencoder for unsupervised anomaly detection. In Proceedings of the IEEE/CVF international conference on computer vision (pp. 1705-1714).
[2] Guo, H., Ren, L., Fu, J., Wang, Y., Zhang, Z., Lan, C., ... & Hou, X. (2023). Template-guided hierarchical feature restoration for anomaly detection. In Proceedings of the IEEE/CVF International Conference on Computer Vision (pp. 6447-6458).

**Questions:**

Please refer to "weakness" section.

---

> ### Author Response · Authors · 2025-12-01
>
> **Q:Your prototype-based reconstruction framework looks quite similar to previous memory / template based reconstruction methods [1] Gong et al. (2019) and [2] Guo et al. (2023), but these works are not cited. Please clarify the novelty over them and properly position your contribution.**
>
>
> We acknowledge that [1] Gong et al. (2019) and [2] Guo et al. (2023) share the high-level concept of utilizing memory/templates to reconstruct normal patterns and detecting anomalies via reconstruction errors. We have cited and discussed these works in the Related Work section of our revised manuscript. In addition,  PIRN significantly differs from [1] and [2] in the following key aspects:
>
> (1) Task setting: [1] and [2] address standard, single-modality 2D anomaly detection and do not consider the few-shot scenario. In contrast, PIRN tackles the more challenging \textit{few-shot Multi-Modal Anomaly Detection (MAD)} task (RGB+3D). Our contributions specifically focus on addressing the difficulties of ``few-shot, multi-modal knowledge fusion, and prototype communication'' under data scarcity.
>
> (2) Methodology: The mechanism for encoding normality in PIRN is different. [1] utilizes explicit external memory slots (an ``explicit memory bank''), while [2] employs template-guided hierarchical restoration using templates from the normal sample library. PIRN introduces a \textit{vector-quantized (VQ) prototype codebook}. Crucially, we introduce Balanced Prototype Assignment (BPA) via optimal transport and Adaptive Prototype Refinement (APR) via GRU. These mechanisms ensure uniform prototype utilization, prevent codebook collapse, and dynamically adapt prototypes to unseen normal variations during testing. This approach differs significantly from the simple memory retrieval or template matching used in the prior works, enabling better stability and coverage in the few-shot setting.
>
>
>
> | **Method**         | **AUROC\_I↑** | **FLOPs↓ (G)** | **Latency↓ (ms)** | **Peak Memory↓ (GB)** |
> |--------------------|--------------|----------------|-------------------|------------------------|
> | M3DM [3]        | 0.845        | 263.27         | 21.61             | **1.26**               |
> | CFM [4]         | 0.845        | 552.53         | 60.57             | 2.35                   |
> | FIND [5]        | 0.921        | 728.46         | 76.09             | 3.28                   |
> | **PIRN (Ours)**    | **0.922**    | **103.36**     | **17.49**         | 1.28                   |
>
>
>
> **Q:  Please clarify whether you have considered the trade-off between complexity and performance.**
>
>
> To clarify the trade-off between complexity and performance, we conducted a detailed efficiency comparison against recent  MAD baselines (M3DM[3], CFM[4], and the recent SOTA, FIND[5]) on the 10-shot MVTec-3D-AD setting. We evaluated accuracy (AUROC$_I$) and efficiency (FLOPs, latency, and Peak GPU Memory) with input $224\times224$ and batch size 1. As summarized in the above table, PIRN achieves the highest AUROC$_I$ (0.922) with remarkable computational efficiency . PIRN requires only 103.36G FLOPs, significantly less than M3DM (263.27G), CFM (552.53G), and FIND (728.46G).
>
> Contrary to the initial impression of complexity, our architectural design choices lead to substantial efficiency gains. Specifically, PIRN achieves SOTA accuracy with **85\% fewer FLOPs** and **4.5 times faster latency** (17.49ms vs 76.09ms) compared to the previous method FIND. This efficiency comes from two key factors. First, unlike methods (CFM, M3DM) that use expensive PointNet-style backbones on raw point clouds, PIRN utilizes efficient 3D encoding by converting point clouds into surface-normal images and applying a 2D ViT (DINO V2). Second, PIRN employs a lightweight decoder. While CFM uses heavy upsampling layers and FIND relies on stacking multiple token-to-token distillation blocks (728.46G FLOPs), PIRN avoids these heavy operations.
>
> Our core modules (BPA, APR, MNC) operate primarily on a compact set of prototypes ($K=10$) rather than all patch tokens ($N=256$). The complexity of these operations (e.g., OT routing, graph attention) is $O(NK + K^2)$, which is negligible compared to the backbone. In conclusion, the experiments demonstrate that PIRN achieves a highly favorable performance-complexity trade-off, offering SOTA accuracy with significantly higher efficiency (lower FLOPs, faster latency) than existing MAD methods.
>
> [3] Wang, Y., Peng, J., Zhang, J., Yi, R., Wang, Y. and Wang, C., 2023. Multimodal industrial anomaly detection via hybrid fusion. CVPR
>
> [4] Costanzino, A., Ramirez, P.Z., Lisanti, G. and Di Stefano, L., 2024. Multimodal industrial anomaly detection by crossmodal feature mapping.CVPR
>
> [5] Li, Y., Liu, F., Liao, J., Tian, S., Foo, C.S. and Yang, X., 2025. FIND: Few-Shot Anomaly Inspection with Normal-Only Multi-Modal Data. ICCV

---

> ### Author Response · Authors · 2025-12-01
>
> \textbf{Q:The paper does not justify why the decoder has 2 layers in few-shot and 8 layers in all-shot settings. Please provide a justification, preferably with an ablation on the number of decoder layers.}
>
>
> | **Layers / 10-shot** | **AUROC\_I** | **AUROC\_P** | **AUPRO** | **F1max\_I** |
> |----------------------|-------------|-------------|-----------|-------------|
> | L = 1                | 0.921       | 0.993       | 0.967     | 0.950       |
> | L = 2                | **0.924**   | **0.994**   | **0.968** | **0.951**   |
> | L = 4                | 0.913       | 0.992       | 0.964     | 0.945       |
> | L = 6                | 0.892       | 0.990       | 0.960     | 0.938       |
> | L = 8                | 0.869       | 0.985       | 0.949     | 0.924       |
>
> | **Layers / All-shot** | **AUROC\_I** | **AUROC\_P** | **AUPRO** | **F1max\_I** |
> |----------------------|-------------|-------------|-----------|-------------|
> | L = 2                | 0.949       | 0.993       | 0.967     | 0.952       |
> | L = 4                | 0.959       | 0.994       | 0.970     | 0.960       |
> | L = 6                | 0.962       | 0.994       | 0.970     | 0.964       |
> | L = 8                | **0.963**   | **0.994**   | **0.973** | **0.966**   |
> | L = 10               | 0.957       | 0.993       | 0.969     | 0.963       |
>
>
> To justify our design choices regarding the decoder depth, we conducted comprehensive ablation studies on the number of decoder layers ($L$) under both the 10-shot  and the all-shot (full data) settings, using only normal data.
>
>
> In the few-shot scenario, the results clearly indicate that a shallower decoder is superior, with $L=2$ providing the optimal trade-off. As shown in the first table, increasing the depth from $L=1$ to $L=2$ improves I-AUROC from 0.921 to 0.924, with corresponding small gains across other metrics. This suggests that a moderate depth can slightly enhance reconstruction capability. However, further increasing the depth to $L=4/6/8$ leads to significant performance degradation. I-AUROC drops sharply from 0.924 ($L=2$) to 0.869 ($L=8$), demonstrating that excessively deep decoders suffer from overfitting in this data-scarce regime. When training data is extremely limited, stacking too many layers causes the model to "memorize" overly specific details of the few normal samples, while also iteratively amplifying noise and estimation errors. Consequently, the model yields high reconstruction errors even for minor normal variations during testing. Thus, the 2-layer decoder strikes the optimal balance, effectively leveraging prototype-based reconstruction and cross-modal communication while avoiding overfitting.
>
>
>
>
> In contrast, the all-shot setting benefits from deeper decoders, with performance saturating around $L=8$. According to the second table, as the number of layers increases from $L=2$ to $L=8$, the I-AUROC steadily improves from 0.949 to 0.963, and AUPRO increases from 0.967 to 0.973. This indicates that with sufficient data, a deeper decoder can progressively enhance performance. With abundant samples, the deep decoder supports multiple iterations of APR, BPA, and MNC, allowing for finer prototype refinement and more thorough exchange of cross-modal normality. However, when the depth is further increased to $L=10$, the performance slightly degrades (I-AUROC drops to 0.957). Excessive depth introduces optimization issues (e.g., over-fitting). Thus, $L=8$ offers the best trade-off between performance and complexity.
>
> Intuitively, few-shot learning requires stronger regularization; a shallow decoder combined with the prototype bottleneck acts as inherent structural regularization. In contrast,  all-shot learning can leverage the increased capacity of a deeper decoder ($L=8$) to fully exploit multi-modal collaboration.
>
> **Q:There is a minor issue: the paper “Revisiting multimodal fusion for 3D anomaly detection from an architectural perspective” appears duplicated in the references.**
>
> We thank the reviewer for pointing this question. The duplicated citation of “Revisiting multimodal fusion for 3D anomaly detection from an architectural perspective” has been removed in the revised manuscript.

---

### Official Review · Reviewer_tAGm · 2025-10-30

**Soundness:** 3
**Presentation:** 3
**Contribution:** 3
**Rating:** 6
**Confidence:** 4

**Summary:**

This paper proposes a multimodal anomaly detection framework that builds on recent prototype-learning ideas. Its core concept is similar in spirit to INP-Former but introduces tailored modules for multimodal settings, especially in modeling cross-modal normality and refining normal prototypes. The authors identify three fundamental challenges in multimodal prototype learning and correspondingly design three key modules. The overall structure is clear, the motivation is sound, and the approach achieves strong performance compared to prior alternatives.

However, the analysis of the proposed modules is relatively shallow. Without deeper insights into why these modules work and how they contribute, the technical credibility and reproducibility are weakened. Moreover, the experimental setup and comparisons contain unclear details that should be clarified for fairness and rigor.

**Strengths:**

The paper explicitly identifies core challenges in multimodal prototype learning and proposes targeted modules to address them. This gives the paper a logical progression.

The proposed components appear reasonable and well-grounded, with links to existing prototype-learning paradigms.

The method outperforms other approaches in the reported benchmarks, suggesting its effectiveness in multimodal anomaly detection.

The model performs well in few-shot scenarios, which is practically valuable for real-world industrial inspection.

**Weaknesses:**

The paper primarily provides quantitative results. There is limited explanation of how or why each module improves performance. This lack of insight weakens the claimed contributions.

The Balanced Prototype Assignment (BPA) appears conceptually similar to coherence loss in INP-Former. Without clearer distinctions, novelty remains questionable.

Since RGB and normal maps capture complementary information, perfect alignment is unrealistic. The paper should justify why alignment is beneficial and how complementary features are preserved.

It is mentioned that INP-Former is trained only on RGB, while other methods in Table 1 seem RGB-D-based. This may make the comparison unfair.

Figure 2 is visually unclear and difficult to follow; Figure 4 offers limited insight and could be condensed.

Eyecandies and MVTec 3D are largely saturated; broader evaluation (e.g., Real-IAD3) would strengthen the experimental evidence.

It is unclear whether this issue occurs only in few-shot cases or in general.

**Questions:**

Can the authors clearly explain how Balanced Prototype Assignment differs from the coherence loss in INP-Former and INP-Former++? Are there theoretical or empirical justifications?

Since RGB and normal maps capture shared and distinct characteristics, what is the motivation for aligning their prototypes? Is there a mechanism to preserve modality-specific cues rather than forcing strict alignment?

In Table 1, is INP-Former trained only with RGB, while others use RGB-D? If so, how is fairness ensured in comparisons?

Can the authors provide deeper interpretation—for example, visualizing the correspondence between learned prototypes and latent features—to explain how prototypes encode normality?

Why does the method generalize so well in the few-shot regime? Which part of the design contributes most?

Does this problem occur only under few-shot settings or also in full-data training?

Will the authors evaluate on Real-IAD3 or other emerging multimodal anomaly detection datasets to further validate effectiveness?

---

> ### Author Response · Authors · 2025-12-01
>
> **Q: Figure 2 is visually unclear and difficult to follow; Figure 4 offers limited insight and could be condensed.**
>
>
> In the revised manuscript, we have completely redesigned this figure (see Sec. 3). The new Fig. 2 presents a cleaner overview of PIRN. We hope the reviewer will find the updated figure in the Method section much clearer. For Fig. 4, we have moved it to the appendix.
>
>
> **Q: “In Table 1, is INP‑Former trained only with RGB, while others use RGB‑D? If so, how is fairness ensured in comparisons?”**
>
> We clarify that the implementation of INP-Former reported in Table 1 is not RGB-only. To ensure a fair comparison in the multi-modal setting, we extended the official 2D INP-Former implementation to handle both RGB and surface-normal inputs. Specifically, we trained two independent INP-Former branches: one for RGB images and the other for surface-normal maps. Crucially, both branches utilized the identical setup as PIRN, including the same ViT-B/14 backbone, number of decoder layers and input resolution.
>
> During inference, each INP-Former branch generates a patch-level anomaly map by computing the discrepancy between encoder and decoder features. These maps are then upsampled and smoothed using a Gaussian filter. We obtain the final fused heatmap by summing the RGB and surface-normal anomaly maps pixel-wise. Therefore, both methods operate on the same multi-modal input information and share an identical scoring procedure. Therefore, the performance advantage of PIRN over this multi-modal INP-Former baseline in Table 1 does not come from utilizing additional modalities or a different evaluation protocol.
>
> **Q: “Can the authors provide deeper interpretation—for example, visualizing the correspondence between learned prototypes and latent features—to explain how prototypes encode normality?”**
>
> To better interpret how PIRN’s prototypes encode normality, we have inserted a new figure in the main paper's experiment section, which visualizes the feature displacement produced by the BPA routing. For several MVTec-3D-AD categories (e.g., bagel, peach) and both RGB and surface-normal branches, we visualize the displacement of patch tokens from their initial feature state ($z_{\text{pre}}$) to their state after BPA+APR+MNC reconstruction ($z_{\text{post}}$). We project prototypes and tokens into a shared 2D PCA space and draw the displacement vectors ($\Delta = z_{\text{post}} - z_{\text{pre}}$). In the plots, gray crosses denote prototypes, translucent lines show per-token movements, and bold arrows indicate the average movement of normal (green) and anomalous (red) tokens.
>
> The visualization reveals a consistent pattern. BPA encourages prototypes to act as stable anchors for distinct normal patterns. Normal tokens start close to prototype clusters and undergo short movements during reconstruction, indicating that the prototype codebook effectively approximates in-distribution patterns. In contrast, anomalous tokens lie farther away and require larger displacements toward normal prototypes. On average, anomalous tokens exhibit 40–50% larger displacement; for example, in the RGB branch of bagel, $|\Delta|{\text{normal}} \approx 6.2$ vs. $|\Delta|{\text{anomaly}} \approx 9.0$. The accompanying displacement histograms further show that normal and anomalous images form almost non-overlapping distributions, with anomalous images consistently shifted to higher $|\Delta|_{2}$ values. This confirms that our prototype-based reconstruction induces strong normal/abnormal contrast at the feature level.

---

> > ### Author Response · Authors · 2025-12-01
> >
> > **Q: Can the authors clearly explain how Balanced Prototype Assignment differs from the coherence loss in INP‑Former and INP‑Former++? Are there theoretical or empirical justifications?**
> >
> > The Balanced Prototype Assignment (BPA) proposed in PIRN differs substantially from the coherence loss employed in INP-Former and INP-Former++ in terms of their optimization objectives and underlying mechanisms. The coherence loss in INP-Former acts primarily as a regularization term. Its objective is to enforce local spatial smoothness within an image and to encourage consistency in token-to-prototype mappings among neighboring patches. However, it does not impose explicit constraints on the overall utilization frequency of individual prototypes.
> >
> > In contrast, BPA (Section 3.2) is formulated as a global balanced Optimal Transport (OT) problem. It is not merely a regularizer but a constrained optimization approach that determines the assignment matrix $T^*$.  BPA enforces strict constraints ensuring two critical conditions: (i) each patch token must distribute a fixed total mass; (ii) crucially, each prototype must receive an approximately equal share of the total assignment mass (uniform utilization). By solving this OT problem, BPA actively enforces a uniform distribution of responsibility among all prototypes at the optimization level. This inherently suppresses codebook collapse and ensures that all prototypes are utilized effectively to capture diverse normal patterns.
> >
> > The mechanisms used to achieve these objectives also differ significantly:
> >
> > (1) Coherence Loss: Coherence loss is typically applied alongside standard, unconstrained assignment mechanisms (e.g., softmax or sigmoid attention). Although the loss penalizes inconsistency, it does not guarantee balanced prototype utilization.
> >
> > (2) BPA and Sinkhorn OT: BPA utilizes the Sinkhorn algorithm to directly solve the constrained OT problem, resulting in the balanced transport plan $T^*$. This mechanism provides a theoretical guarantee (inherent to balanced OT) that prototype usage is uniform rather than overly focusing on a few prototypes.
> >
> > We provide strong empirical evidence supporting the need for an explicit balanced assignment in Table 3 . In this ablation study, we compared BPA (Balanced OT) with alternative assignment mechanisms such as Softmax, Linear, and Sigmoid attention. These alternatives can be viewed as assignment strategies lacking explicit balancing constraints. The results demonstrate a significant performance gap: BPA achieves 92.2% I-AUROC, significantly outperforming Softmax (83.2%), Linear (84.5%), and Sigmoid (87.8%) attention.
> >
> > **Q: Since RGB and normal maps capture shared and distinct characteristics, what is the motivation for aligning their prototypes? Is there a mechanism to preserve modality-specific cues rather than forcing strict alignment?**
> >
> > We would like to clarify that the goal of MNC is not to enforce strict feature-level alignment between RGB and normal maps. Unlike methods such as CFM that pursue dense, pixel-wise cross-modal matching, MNC performs only loose, prototype-level alignment to facilitate the exchange of high-level normality concepts, rather than fine-grained modality details (Section 3.4).
> >
> > In MNC Stage 1 (Prototype Alignment), the motivation is to establish a shared understanding of high-level normal concepts (e.g., "flat surface," "sharp edge"). We utilize a Graph Attention Network (GAT) constructed over a K-Nearest Neighbors (KNN) graph structure. The KNN structure ensures that edges are only established between prototypes that are semantically similar across modalities. Prototypes that capture highly modality-specific patterns (e.g., complex color variations without geometric change) will have few or no connections to the other modality. These prototypes retain their unique information. Furthermore, each modality maintains its distinct prototype set ($P_{rgb}, P_{sn}$).
> >
> > In MNC Stage 2 (Cross-Modal Normality Injection), this stage injects cross-modal information while actively preventing the suppression of modality-specific cues. We use the purified patch tokens of the current modality as queries and the prototypes of the other modality as keys/values. If the geometric and textural information are inconsistent for a given patch, the attention weights will naturally be low. Crucially, we introduce a learnable gate $g$ (Eq. 5) to dynamically modulate the influence of the cross-attention output. This allows the network to suppress unreliable cross-modal injection, ensuring that information is supplemented only where semantic consistency exists.
> >
> > The effectiveness of this approach in preserving complementarity is demonstrated in Fig. 5 (appendix). We observe that the RGB branch excels at detecting pure texture anomalies (Fig. 5.1), while the surface normal branch is more sensitive to shape deformations (Fig. 5.2). This confirms that MNC does not force the two modalities to become identical.

---

> > > ### Author Response · Authors · 2025-12-01
> > >
> > > **Q: Why does the method generalize so well in the few-shot regime? Which part of the design contributes the most?**
> > >
> > > We provide a quantitative decomposition based on the ablation study (Tab. 2) and a conceptual analysis to explain PIRN's strong generalization in the few-shot regime. The baseline model, a vanilla prototype-based reconstruction framework (without BPA, APR, or MNC), achieves only 82.8% I-AUROC. This significantly lower performance confirms that the naive prototype-based reconstruction struggles to capture diverse normal patterns when data is scarce.
> > >
> > > The most significant contribution comes from the Balanced Prototype Assignment (BPA). Adding BPA to the baseline yields the greatest performance improvement, increasing I-AUROC by +5.5% (to 88.3%). This shows that enforcing balanced utilization of prototypes (Fig. 1 Right) is critical in the few-shot regime. With limited normal samples, BPA effectively mitigates codebook collapse by ensuring a diverse representation of normality.
> > >
> > > Conceptually, the robustness of PIRN is based on two key design principles. First, the Information Bottleneck via Compact Prototypes is crucial. Unlike memory-bank methods that store numerous feature exemplars, PIRN utilizes a compact, finite set of learnable prototypes. This design retains only essential normal patterns, making the model inherently less prone to overfitting the limited training data. Second, the Expanded Normality Coverage provided by the OT-based BPA mechanism forces the model to utilize all available prototypes uniformly. This maximizes the diversity of normal patterns captured by the codebook, ensuring broader coverage of the “normal space”, rather than collapsing onto a few dominant modes learned from the few available examples.
> > >
> > > **Q: Does this problem occur only under few-shot settings or also in full-data training?**
> > >
> > > The authors thank the reviewer for this pertinent question. By “this problem,” we refer to the limitations inherent in conventional alignment-based and memory-based methods: specifically, their susceptibility to misidentifying unseen normal variations as anomalies. This occurs either due to unreliable cross-modal correspondences or insufficient coverage of the true normal distribution when the training data lack diversity. These issues occur in both settings but are significantly exacerbated in the few-shot regime.
> > >
> > > When training samples are scarce, existing methods struggle to capture the full spectrum of normality, leading to a substantial generalization gap. PIRN addresses this directly by promoting diverse representations (BPA) and adapting to novel variations (APR). As evidenced in Tab. 1, this yields significant performance improvements. For example, in the 10-shot setting, PIRN outperforms the strongest baseline by a large margin of +3.7% (MVTec-3D AD) and +4.0% (Eyecandies) in I-AUROC.
> > >
> > > In the full-data regime, access to abundant normal samples allows the other methods to partially mitigate these issues, leading to near-saturated performance across different approaches. However, the underlying challenge (e.g., prototype collapse) still exists. Although the performance differences are smaller, PIRN still maintains an advantage. As shown in Tab. 1, PIRN achieves the highest mean I-AUROC of 0.963 on MVTec-3D-AD, surpassing the closest competitors (e.g., CFM at 0.954).

---

> ### Author Response · Authors · 2025-12-03
>
> **Will the authors evaluate on Real-IAD3 dataset to further validate effectiveness? **
>
> | Modality | RGB–Cflow I-AUROC | RGB–Cflow P-AUROC | RGB–SimpleNet I-AUROC | RGB–SimpleNet P-AUROC | 3D–PointMAE I-AUROC | 3D–PointMAE P-AUROC | 2D+3D–AST I-AUROC | 2D+3D–AST P-AUROC | 2D+3D–PointMAE+PatchCore I-AUROC | 2D+3D–PointMAE+PatchCore P-AUROC | 2D+3D–M3DM I-AUROC | 2D+3D–M3DM P-AUROC | D^3–D^3M I-AUROC | D^3–D^3M P-AUROC | RGB+SN–PIRN (Ours) I-AUROC | RGB+SN–PIRN (Ours) P-AUROC |
> |---|---|---|---|---|---|---|---|---|---|---|---|---|---|---|---|---|
> | `audio_jack_socket` | 0.943 | _0.944_ | 0.973 | 0.926 | 0.763 | 0.655 | 0.860 | 0.590 | 0.926 | 0.673 | _0.981_ | 0.699 | **0.983** | 0.757 | 0.950 | **0.964** |
> | `common_mode_filter` | 0.271 | 0.847 | 0.717 | 0.822 | 0.725 | 0.687 | **0.899** | 0.802 | 0.523 | 0.922 | 0.580 | _0.934_ | 0.618 | **0.947** | _0.826_ | 0.883 |
> | `connector_housing-female` | 0.839 | 0.921 | 0.795 | 0.891 | _0.958_ | 0.428 | 0.914 | 0.716 | 0.870 | 0.919 | 0.920 | **0.979** | 0.931 | 0.951 | **0.972** | _0.971_ |
> | `crimp_st_cable_mount_box` | 0.18 | 0.442 | 0.372 | 0.745 | 0.291 | 0.363 | 0.485 | 0.589 | _0.713_ | 0.931 | _0.749_ | 0.933 | **0.811** | **0.969** | 0.659 | _0.961_ |
> | `dc_power_connector` | 0.661 | 0.726 | 0.661 | 0.725 | 0.849 | 0.507 | **0.995** | 0.770 | 0.720 | 0.921 | 0.715 | _0.950_ | 0.922 | 0.947 | _0.944_ | **0.994** |
> | `ethernet_connector` | 0.967 | 0.853 | 0.981 | 0.866 | **1** | 0.656 | _1.000_ | 0.906 | 0.947 | 0.956 | 0.983 | _0.978_ | 0.996 | 0.970 | 0.997 | **0.992** |
> | `ferrite_bead` | 0.529 | 0.914 | 0.408 | 0.806 | 0.634 | 0.717 | 0.894 | 0.817 | 0.913 | 0.932 | _0.965_ | 0.966 | **0.967** | _0.978_ | 0.717 | **0.993** |
> | `fork_crimp_terminal` | 0.462 | 0.657 | 0.416 | 0.945 | 0.422 | 0.62 | 0.595 | 0.773 | 0.769 | 0.952 | 0.780 | _0.964_ | _0.819_ | 0.946 | **0.978** | **0.991** |
> | `fuse_holder` | 0.853 | 0.861 | 0.564 | _0.957_ | 0.309 | 0.605 | 0.597 | 0.754 | 0.736 | 0.927 | 0.770 | 0.948 | 0.866 | 0.915 | **0.998** | **0.996** |
> | `headphone_jack_socket` | **0.996** | 0.914 | 0.933 | 0.879 | 0.607 | 0.633 | 0.660 | 0.696 | 0.919 | 0.942 | _0.982_ | 0.982 | _0.994_ | **0.987** | 0.942 | 0.975 |
> | `humidity_sensor` | **0.781** | 0.836 | 0.737 | 0.89 | 0.644 | 0.562 | 0.565 | 0.723 | 0.689 | 0.933 | 0.717 | _0.958_ | _0.78_ | **0.969** | 0.744 | **0.991** |
> | `knob_cap` | 0.637 | 0.893 | 0.672 | 0.879 | 0.656 | 0.425 | 0.919 | 0.656 | 0.903 | _0.958_ | _0.925_ | 0.938 | **0.931** | 0.947 | 0.923 | **0.976** |
> | `lattice_block_plug` | 0.833 | 0.852 | 0.79 | 0.898 | 0.769 | 0.776 | 0.842 | 0.919 | 0.911 | 0.923 | _0.917_ | 0.958 | **0.939** | 0.941 | 0.892 | **0.969** |
> | `lego_pin_connector_plate` | 0.828 | 0.877 | 0.857 | _0.947_ | 0.361 | 0.482 | 0.847 | 0.629 | 0.662 | 0.759 | 0.681 | 0.734 | 0.891 | 0.889 | **0.981** | **0.980** |
> | `lego_propeller` | 0.615 | 0.739 | _0.939_ | 0.799 | 0.348 | 0.62 | 0.471 | 0.703 | 0.540 | 0.727 | 0.530 | 0.773 | 0.739 | _0.863_ | **1.000** | **0.933** |
> | `limit_switch` | 0.846 | **0.95** | 0.823 | 0.79 | 0.763 | 0.545 | 0.804 | 0.641 | 0.822 | 0.938 | 0.863 | **0.966** | _0.925_ | **0.984** | **0.961** | _0.971_ |
> | `miniature_lifting_motor` | 0.402 | 0.799 | 0.402 | 0.76 | 0.717 | 0.435 | 0.766 | 0.467 | _0.948_ | _0.962_ | **0.975** | **0.991** | 0.823 | 0.961 | 0.604 | 0.838 |
> | `power_jack` | 0.354 | 0.664 | 0.176 | 0.489 | 0.433 | 0.687 | 0.564 | 0.645 | _0.981_ | 0.923 | **0.996** | 0.902 | 0.973 | **0.947** | 0.595 | 0.862 |
> | `purple_clay_pot` | 0.343 | 0.571 | 0.343 | 0.938 | 0.869 | 0.271 | 0.635 | 0.445 | 0.921 | _0.961_ | _0.944_ | 0.953 | **0.962** | 0.922 | 0.871 | **0.997** |
> | `telephone_spring_switch` | 0.575 | 0.91 | 0.627 | 0.916 | 0.771 | 0.413 | **0.951** | 0.551 | 0.827 | 0.944 | 0.856 | 0.936 | _0.934_ | _0.957_ | 0.904 | **0.987** |
> | **Avg** | 0.645 | 0.808 | 0.659 | 0.843 | 0.644 | 0.554 | 0.693 | 0.650 | 0.812 | 0.905 | 0.841 | 0.922 | **0.890** | _0.937_ | _0.873_ | **0.961** |
>
> We conducte comprehensive experiments on the challenging Real-IAD D3 dataset in the all-shot normal training setting and added them in main paper. The experimental results are shown in the above table. Overall, PIRN achieves  the best overall anomaly localization (P-AUROC) of 0.961 and the second-best overall anomaly detection (I-AUROC) of 0.873.
>
> In particular, PIRN demonstrates outstanding localization capabilities, achieving an average P-AUROC of 0.961 over the D³M baseline (0.937). Notably, PIRN achieves the highest P-AUROC in 13 out of the 20 categories.  In terms of I-AUROC, PIRN achieves a strong score of 0.873, closely following D³M (0.890).  D³M is specifically designed to leverage the unique D³ data representation (combining RGB, Pseudo-3D, and 3D inputs). In contrast, PIRN operates using only two standard modalities: RGB and Surface Normals (RGB + SN). Despite utilizing a simpler input representation, PIRN maintains competitive detection rates while delivering superior localization accuracy.

---

### Meta-Review · Area_Chair_CkDU · 2026-01-06

**Summary:**

This paper proposes PIRN, a prototype-driven reconstruction framework for few-shot multimodal anomaly detection. Reviewers generally agree that the paper addresses an important and practical problem, particularly the performance degradation of existing multimodal anomaly detection methods under data-scarce settings. The core design is viewed as well-motivated and logically structured, and the method demonstrates strong empirical performance on standard benchmarks. While some reviewers initially viewed the contribution as incremental or borderline, the overall tone was cautiously positive, with multiple reviewers indicating they would not object to acceptance.

**Reviewer Concerns:**

The concerns addressed by the rebuttal are listed as follows.
1. The authors explicitly position PIRN against earlier memory/template-based methods (e.g., Gong et al. 2019; Guo et al. 2023), clarifying key differences in task setting and methodology.
2. The rebuttal adds detailed visualizations and quantitative analyses that provide clearer intuition on how BPA anchors normal patterns and how anomalies lead to larger feature displacements.
3. The INP-Former baseline is clarified and re-implemented in a multi-modal setting with identical backbones, training splits, and scoring pipelines, addressing concerns about unfair RGB-only comparisons.
4. Extensive efficiency analysis (FLOPs, latency, and  training time) demonstrates that PIRN achieves competitive or superior accuracy with  lower computational cost.

The remaining concerns are listed as follows.
1. Some reviewers still view the overall contribution as incremental relative to existing prototype-learning paradigms.
2. The framework is architecturally more complex than earlier reconstruction-based methods.

Overall, the major concerns raised during review have been  addressed.

**Reviewer Scores:**

The AC thinks that most reviewers will maintain their scores or slightly increase them, as the authors provided additional clarifications or experiments, and partially resolved several of the key concerns raised during the review process.

---

### Decision · Program_Chairs · 2026-01-26

Accept (Poster)